# Updating temporal expectancy of an aversive event engages striatal plasticity under amygdala control

Glenn Dallérac[1],[*],[†], Michael Graupner[2],[*],[†], Jeroen Knippenberg[1], Raquel Chacon Ruiz Martinez[3], Tatiane Ferreira Tavares[1], Lucille Tallot[1], Nicole El Massioui[1], Anna Verschueren[1],[4], Sophie Höhn[1], Julie Boulanger Bertolus[1],[5], Alex Reyes[2], Joseph E. LeDoux[2],[6], Glenn E. Schafe[7], Lorenzo Diaz-Mataix[2] & Valérie Doyère[1]

Pavlovian aversive conditioning requires learning of the association between a conditioned stimulus (CS) and an unconditioned, aversive stimulus (US) but also involves encoding the time interval between the two stimuli. The neurobiological bases of this time interval learning are unknown. Here, we show that in rats, the dorsal striatum and basal amygdala belong to a common functional network underlying temporal expectancy and learning of a CS–US interval. Importantly, changes in coherence between striatum and amygdala local field potentials (LFPs) were found to couple these structures during interval estimation within the lower range of the theta rhythm (3–6 Hz). Strikingly, we also show that a change to the CS–US time interval results in long-term changes in cortico-striatal synaptic efficacy under the control of the amygdala. Collectively, this study reveals physiological correlates of plasticity mechanisms of interval timing that take place in the striatum and are regulated by the amygdala.

[1] Institut des Neurosciences Paris-Saclay (Neuro-PSI), Cognition and Behaviour Department, UMR 9197, Université Paris Sud, CNRS, Université Paris Saclay, Orsay F-91405, France. [2] Center for Neural Science, New York University, New York, New York 10003, USA. [3] Laboratory of Neuromodulation, Teaching and Research Institute, Hospital Sirio Libanes, Rua Professor Daher Cutait, 69, Sao Paulo 01308-060, Brazil. [4] École Normale Supérieure, Paris F-75005, France. [5] École Normale Supérieure, Lyon F-69007, France. [6] Nathan Kline Institute for Psychiatric Research, Orangeburg, New York 10962, USA. [7] Department of Psychology, Hunter College, New York, New York 10065, USA. * These authors contributed equally to this work. † Present addresses: Collège de France, CIRB, CNRS UMR 7241, INSERM U1050, Paris F-75005, France (G.D.); Laboratoire de Physiologie Cérébrale, UMR 8118, Université Paris Descartes, Paris 75006, France (M.G.). Correspondence and requests for materials should be addressed to V.D. (email: valerie.doyere@u-psud.fr).

L earning temporal relationships between events enables organisms to build predictions and develop adaptive behaviour accordingly. In associative learning, subjects not only learn the association but also the temporal contingencies between the stimuli. Pavlovian aversive conditioning is one of the most widely used learning paradigms in neuroscience and has advanced our understanding of the neural mechanisms of associative learning[1]. In this paradigm, a neutral stimulus, the conditioned stimulus (CS), acquires a predictive value for an unconditioned aversive stimulus (US) that has an inherent value. The emergence of behavioural and physiological correlates of temporal expectancy of the US during learning, observed in humans and other animals, demonstrates that subjects encode the time interval between the two stimuli. For instance, rats typically show maximal levels of fear-potentiated startle, and changes in heart and respiration rate at the expected time of the shock US[2–4]. Although such a temporal aspect has been suggested to be a fundamental component of associative learning[5], its neurobiological basis remains poorly understood.

One dominant model for temporal processing is the striatal beat-frequency (SBF) in which medium spiny striatal neurons integrate cortical oscillatory patterns of activity and act as coincidence detectors when an aversive or appetitive US is presented[6,7]. A primary assumption of this model is that striatal inputs, in particular afferents from the prefrontal cortex, are continuously updated in a way that allows for the scalar property (that is, temporal precision proportional to the timed interval), a fundamental feature of interval timing[8]. Hebbian plasticity mechanisms, including long-term potentiation and long-term depression (LTD), are proposed to underlie the storage of reference coincidence patterns. To date, the most compelling electrophysiological evidence come from two studies[9,10] showing that firing of neuronal ensembles in the dorsal striatum follows the behaviourally measured temporal expectancy of food availability. Neuroimaging investigations of interval timing in humans and studies in animals have implicated multiple brain regions, and in particular the dorsal striatum and prefrontal cortex[11].

Among the multiple brain regions implicated in temporal processing, a growing body of evidence points to the amygdala as a potential player in timing the CS–US interval[12]. We have recently observed that a simple change in the arrival time of the US triggers plasticity mechanisms in the lateral amygdala during Pavlovian aversive conditioning[13]. Several studies have observed that neuronal activity of different amygdala nuclei markedly increases slightly before US presentation[14–16]. Although such observations suggest that the amygdala plays a role in temporal expectancy of the aversive event, the protocols used in the latter investigations were not designed to address the timing processes *per se* and thus do not rule out other potential causes of changes in neuronal activity such as motor activity or the associative component of learning. Hence, whether and how the amygdala is involved in interval timing remains unknown. Interestingly, there are direct amygdala projections to the striatum[17], providing an anatomical substrate for functional interactions for processing the CS–US interval.

In the current study, we asked whether the dorsal striatum forms, with the amygdala, a functional network that is at play in temporal expectancy of an aversive US and whether these structures undergo neural changes when the animal learns a new CS–US interval. To do so, we developed an experimental paradigm using auditory aversive conditioning in which the time from CS onset is the only predictor of the US arrival[13,18,19]. In this protocol, non-reinforced probe trials and a shift in CS–US interval allow us to isolate the temporal aspect of US expectancy and its scalar property. Using this paradigm, we unravel that

temporal expectancy implies a network where the coherence between the dorsal striatum and amygdala is at play. In line with this, we find that updating the CS–US time interval induces long-term changes in cortico-striatal synaptic efficacy under the control of the amygdala.

## Results

**Neural correlates of temporal expectancy.** We recorded dorsal striatum and amygdala local field potentials (LFP) in rats performing a task that involves processing a CS–US time interval, in which the tone (CS) extends beyond the arrival of the US (footshock) and thus time from CS-onset is the sole predictor of US arrival (Fig. 1a). Rats were trained for several weeks to lever-press for food and subjected to a tone-shock aversive conditioning protocol while lever-pressing with a CS–US interval of 30 s for more than 60 sessions. They were then implanted with electrodes into the striatum and amygdala, and retrained for at least 10 sessions. These well-trained rats showed a bell-shaped curve of lever-pressing suppression on non-reinforced probe trials, typical of a temporal expectancy for US arrival. The maximum conditioned suppression was at a time close to the US arrival, although anticipatory (average peak time at $22.5 \pm 0.9$ s; Fig. 1b), confirming previous reports using similar procedures[19,20]. Furthermore, shifting the CS–US interval from 30 s to 10 s yielded an immediate shift in the peak of suppression (before shift versus 1st session of shift: time X session interaction, $F_{59,295} = 1.94$, $P < 0.001$, Fig. 1b) leveling off at a proportional reduction in peak time ($8.6 \pm 0.7$ s) within 5 sessions, while keeping the number of lever-pressing during the inter-trial intervals at a stable level ($1.16 \pm 0.18$ to $0.98 \pm 0.20$ lever-press per second; $P = 0.07$). Once the behaviour had adapted to the new CS–US interval, the width of the suppression curve was also reduced accordingly. There was good superposition of the pre- and post-shift suppression curves when plotted on normalized axes (high $\eta^2$ value, an index of superposition, $\eta^2 = 0.920$, and no before versus after shift, time X session interaction, $F_{19,95} = 1.06$, $P = 0.40$), as predicted by the scalar property of interval timing (Fig. 1c). Furthermore, calculation of the Pearson correlation coefficient showed that positive correlations between the 30 s versus 10 s curves were only significant after normalization of time (Supplementary Table 1). Thus, the shift of peak time was accompanied by a corresponding change in the width of the suppression curves, in agreement with the scalar property. Therefore, our protocol allows us to assess the scalar property for potential neural correlates, as well as to assess whether plasticity mechanisms underlie fast learning of a new CS–US interval once all the other contingencies have been learned.

To test whether the striatum and the amygdala act together in a real-time neural network to process interval timing in this aversive associative task, we recorded LFP in dorsomedial striatum (DMS) and basal nucleus of the amygdala (BA, Fig. 2a) in interleaved sessions during which animals did not have access to the lever, thus reducing movement-related artifacts and dissociating electrophysiological activity from changes in motor control. We chose to record from these two regions because (1) DMS, unlike dorsolateral striatum (DLS), has been found to be associated with expectancy and flexibility following changes in task contingencies in reward settings[21–23], and (2) there is a direct projection from the BA to the DMS, which has been suggested to have a potential role in interval timing[12,24]. When normalized to pre-CS baselines, which were similar for 30 and 10 s conditions (Supplementary Fig. 1), the relative power spectral density (PSD) revealed changes in oscillations during the CS in the theta frequency band with two delimited sub-bands corresponding to low- (3–6 Hz) and high- (6–9 Hz) theta

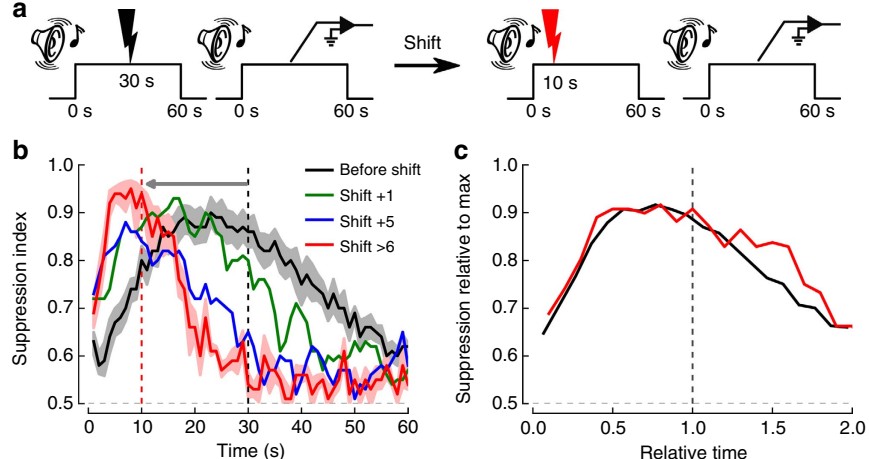

**Figure 1 | Conditioned suppression shows temporal expectancy that follows the scalar property after a change of the CS–US interval.** (**a**) Graphical representation of the paradigm used to investigate correlates of interval timing. Note that the auditory CS does not co-terminate with the US arrival, thus enabling to isolate the temporal component of learning once all other contingencies have been acquired. Recordings during CS alone trials were analysed for interleaved sessions without access to lever. (**b**) Suppression index during CS presentation. Lever-pressing suppression increases from baseline (0.5) during the CS. Learning of the new time of US interval (10 s) occurs rapidly as the suppression curve shifted within one training day (black versus green curves) and stabilized at day 5 (blue and red curves). For clarity, s.e.m. are shown for the 'before shift' ($n = 8$) and 'shift $> 6$' curves only ($n = 6$). (**c**) Suppression index for 30 and 10 s expected CS–US intervals on normalized axes. Relative time refers to normalization of time to the occurrence of the actual time of US arrival. To assess the scalar property without the confound of peak rate differences, both curves were also normalized to their respective maximal values. As predicted by the scalar property, the width of the curve was reduced in accordance with the time shift, as there was good superposition of both curves.

rhythm[25–27] (Fig. 2b heat plots in the upper panel), the lower band showing the strongest increase in power centred $\sim 30$ s (the expected time of US arrival). The strongest increase in PSD moved to 10 s when the time of US arrival was shifted to 10 s (Fig. 2c heat plots in the upper panel). Non-parametric cluster analysis on per-rat PSD averages revealed an onset response at the beginning of the CS in both the 30 and 10 s conditions in the DMS but not in BA (Fig. 2b,c, lower panels). A significant PSD increase was observed in the DMS, in particular in the 3–6 Hz band, during almost the entire CS duration when the US was expected at 30 s (Fig. 2b, lower panel), and for a shorter duration in the 10 s condition (Fig. 2c, lower panel). To test whether these increases are related to the processing of the CS–US time interval, a two-way analysis of variance (ANOVA) was performed on per-rat PSD averages within these specific frequency sub-bands (Fig. 2d). A significant interaction between condition (US@30 versus US@10) and elapsed time would indicate that the time course of the changes is affected by the CS–US interval. The ANOVA analyses revealed a significant time X condition interaction between the non-shifted (30 s) and shifted (10 s) conditions in the 3–6 Hz band in the DMS ($P < 0.001$; all statistics related to LFP analyses are reported in Supplementary Table 2), thereby implicating the striatal low theta rhythm in CS–US interval timing. Although the changes were fairly modest in amplitude, there was also a significant interaction in the BA ($P < 0.05$). In contrast, the PSD analysis in the 6–9 Hz band showed no significant time X condition interaction, in either the DMS or the BA. We also analysed gamma frequencies following the same strategy (Fig. 3). Cluster-based non-parametric statistics revealed significant increases in gamma power in both DMS and BA when the US was expected at 30 s and at 10 s (Fig. 3a,b). Narrowing PSD analysis to the 60–70 Hz band also revealed significant time X condition (US@30 versus US@10) interactions in both structures (Fig. 3c; Ps $< 0.01$). In sum, both the striatum and the BA showed a timing-related increase in PSD, specifically in the 3–6 Hz low theta range and in the 60–70 Hz gamma range.

In order to evaluate whether DMS and BA may form a functional network with regard to interval timing processes, we then calculated the coherence between the LFP signals from both structures, which quantifies synchronized oscillations as a function of frequency. The coherence did not show an onset response and the largest significant increases in the 3–6 Hz and the 6–9 Hz frequency bands occurred around the expected time of US arrival in the 30 s and the 10 s conditions (Fig. 2e,f, upper and lower panels). Coherent oscillations in the 3–6 Hz band showed a significant time X condition (US@30 versus US@10) interaction ($P < 0.01$, Fig. 2g). In contrast, coherence analysis in the 6–9 Hz theta band or in the gamma (60–70 Hz) band did not show a significant time X condition interaction (Fig. 3d,e,f). These data show that the coherence between DMS and BA oscillations in the 3–6 Hz range dynamically increases while processing the CS–US interval in relation to the expected time of US arrival. To control for the effect of the 60 s tone itself on LFP, we analysed theta and gamma bands in both DMS and BA during 2 consecutive days of CS exposure without US delivery in naive rats, and found no significant variation in PSD and coherence (Supplementary Fig. 2), thus confirming that the variation in LFP oscillations in conditioned animals were attributable to expectation of US arrival.

Thus far, our results clearly show that the PSD in DMS and BA, as well as the coherence between these two structures, follow the US arrival time (US@30 and US@10) with a time course that adapts both its maximum and width in the averaged curves (Figs 2d,g and 3c), strongly suggesting they follow the scalar property, as does behaviour (Fig. 1c). We thus tested which of these neural correlates better follow the scalar property of timing by rescaling US@30 and US@10 PSD and coherence according to the US arrival time (Fig. 4). Normalization of all axes revealed good superposition of all curves, except for the BA 3–6 Hz band, and also clearly highlighted the embedded onset response in the PSD (Fig. 4a,b,d,e). In agreement all but the BA low theta PSD interactions were lost while a significant time effect remained (Ps $< 0.05$; Supplementary Table 2), thus confirming the lock of

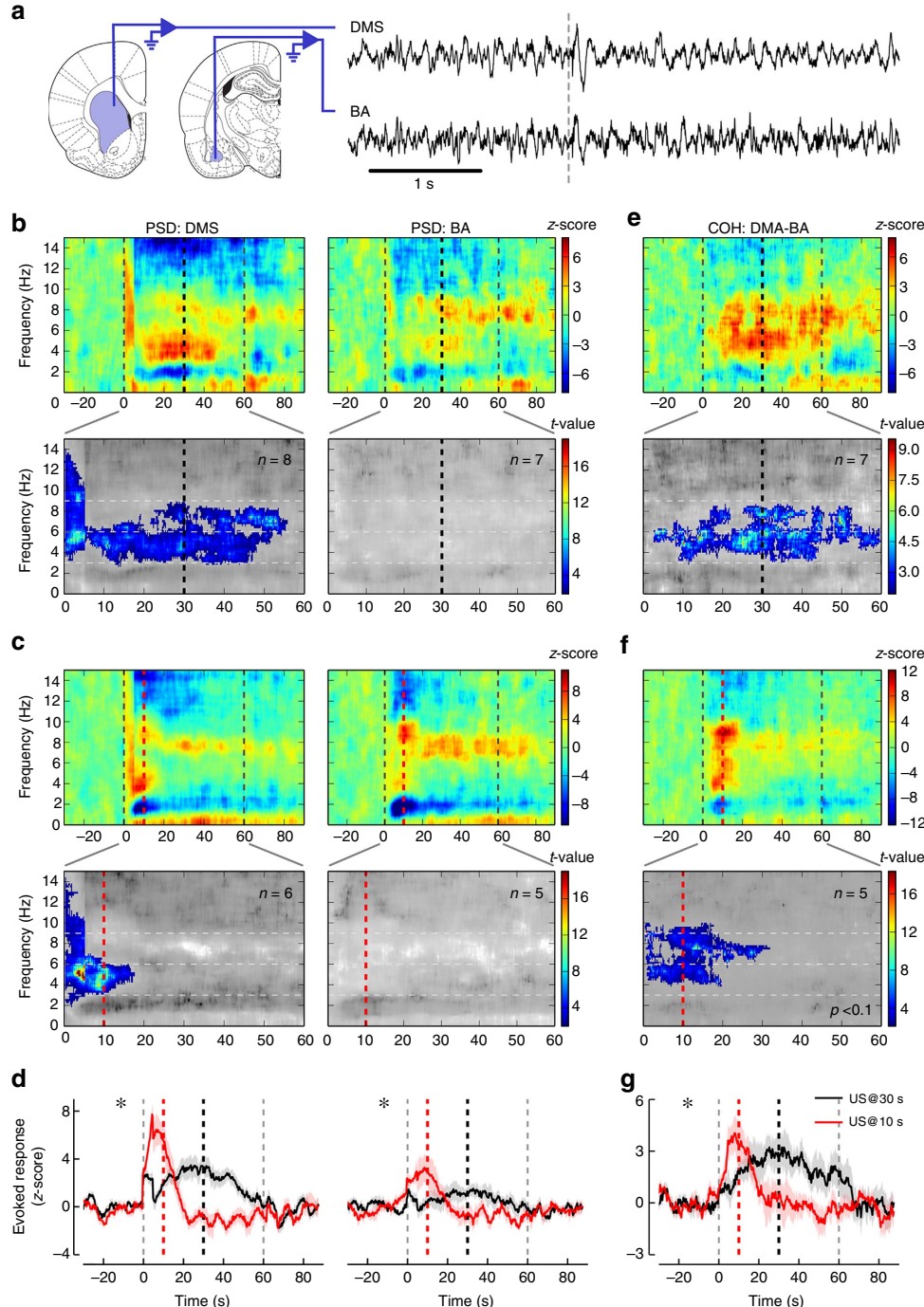

**Figure 2 | Time-related oscillatory changes in the 3–6 Hz frequency band recorded from DMS and BA during CS alone presentations.** (**a**) Sketch of the recording locations for dorso-medial striatum (DMS, left) and basal amygdala (BA, middle), adapted, with permission, from drawings published in ref. 64. Raw traces of the simultaneously recorded local field potentials in the DMS (right, upper trace) and the BA (right, lower trace) before and during CS presentation (onset marked by dashed grey line). (**b**) Relative change in LFP power for the DMS (upper left) and the BA (upper right) before, during and after 60 s CS presentation (onset and offset marked by dashed grey lines) when the US is expected 30 s after the CS onset (thick black dashed line). The relative change with respect to pre-stimulus baseline per frequency shows increased activity in two bands (3–6 Hz and 6–9 Hz). Non-parametric cluster analysis (lower panels) reveals significant ($P < 0.05$) power spectrum increases for the DMS but not for the BA during CS presentation. The grey colour code depicts non-significant changes as $t$-values. (**c**) Relative change in LFP power for the DMS (upper left) and the BA (upper right) before, during and after CS presentation after the US arrival was shifted to 10 s (thick red dashed line). Same depiction as in **b**. (**d**) Comparison of the mean ± s.e.m. changes in LFP power in the 3–6 Hz band between 30 s (black) and 10 s (red) expected time of US arrival conditions (DMS: left panel; BA: right panel). Stars mark significant time X condition interaction (two-way ANOVA with Geisser and Greenhouse correction; $P < 0.001$ and $P = 0.01$, respectively).
(**e**) Relative change in LFP coherence between DMS and BA in the 30 s condition (upper panel). The relative change with respect to pre-stimulus baseline (60 s) per frequency shows significant increases in the theta range during the stimulus presentation (lower panel). (**f**) Relative change in LFP coherence between DMS and BA after the US arrival was shifted to 10 s. Same depiction as in **e**. (**g**) Comparison of the relative coherence dynamics in the 3–6 Hz frequency band between 30 s (black) and 10 s (red) expected US arrival conditions. The star marks significant time X condition interaction (two-way ANOVA with Geisser and Greenhouse correction; $P < 0.01$).

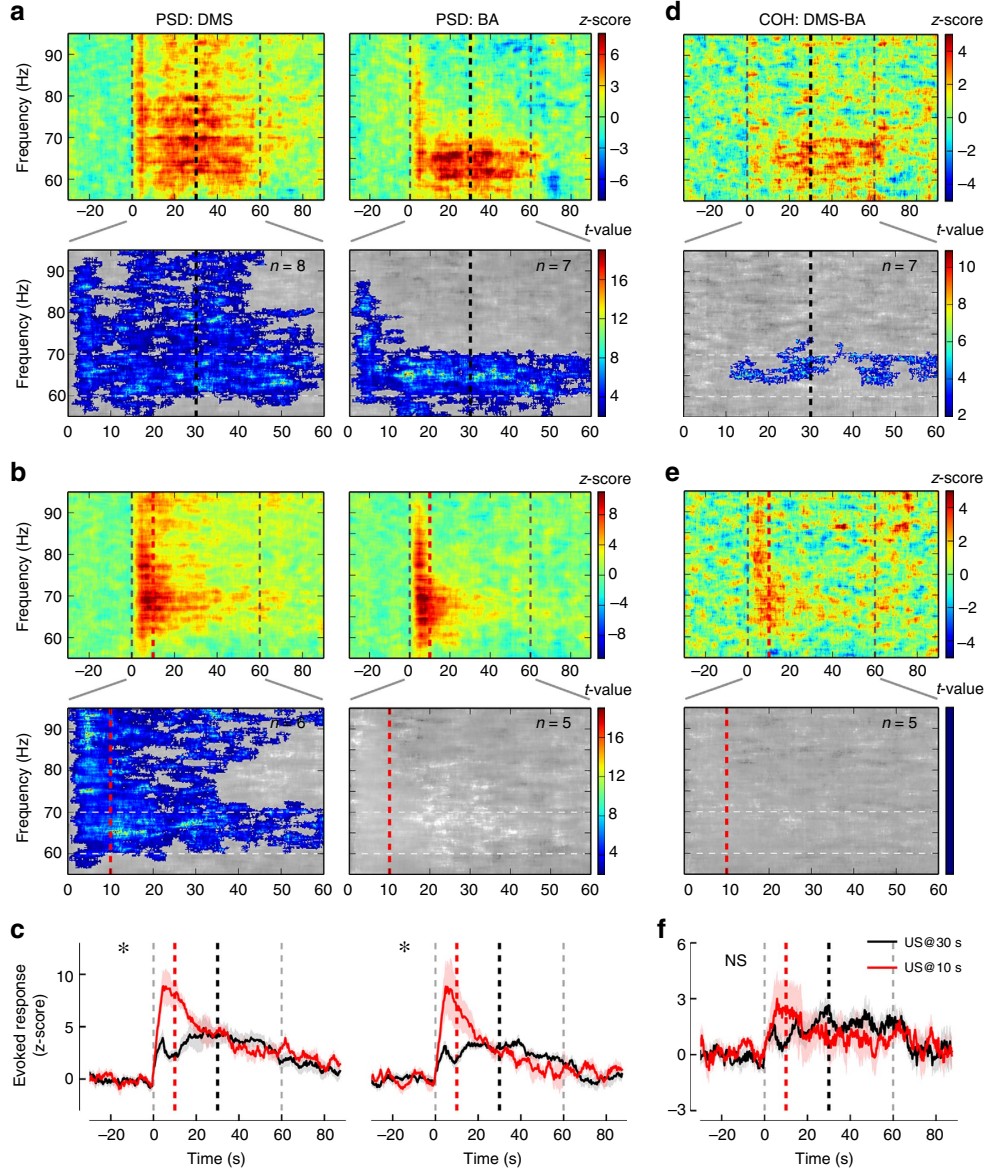

**Figure 3 | Time-related oscillatory changes in the 60–70 Hz frequency band recorded from DMS and BA during CS alone presentations.** (**a**) Relative change in LFP power for the DMS (upper left) and the BA (upper right) before, during and after 60 s CS presentation (onset and offset marked by dashed grey lines) when the US is expected 30 s after the CS onset (thick black dashed line). The relative change with respect to pre-stimulus baseline per frequency shows increased activity in a the 60–95 Hz range in the DMS and in the 60–70 Hz range in the BA. Non-parametric cluster analysis (lower panels) reveals significant ($P < 0.05$) power spectrum increases for the DMS and the BA during the CS presentation. The grey colour code depicts non-significant changes as $t$-values. Expected time of US arrival at 30 s is shown by a thick black dashed line. (**b**) Relative change in LFP power for the DMS (upper left) and the BA (upper right) before, during and after CS presentation after the US arrival was shifted to 10 s (thick red dashed line). Same depiction as in **a**. Significant increases in the LFP power are observed in the DMS (lower left) but not in the BA (lower right). (**c**) Comparison of the mean ± s.e.m. change in LFP power in the 60–70 Hz frequency band between 30 s (black) and 10 s (red) expected time of US arrival conditions (DMS: left panel; BA: right panel). Stars mark significant time X condition interaction (two-way ANOVA with Geisser and Greenhouse correction; both $P < 0.01$). (**d**) Relative change in LFP coherence between DMS and BA before, during and after CS presentation in the 30 s condition (upper panel). Cluster analysis (lower panel). (**e**) Relative change in LFP coherence between DMS and BA after the US arrival was shifted to 10 s. Same depiction as in **d**. No significant coherence increases occur during stimulus presentation (see lower panel). (**f**) Comparison of the relative coherence dynamics in the 60–70 Hz frequency band between 30 s (black) and 10 s (red) US arrival conditions.

changes in the oscillations to time estimation. Comparison of the superposition index ($\eta^2$) between averaged curves indicated that the highest superposition was observed for 3–6 Hz coherence (Fig. 4c) as compared to the PSD frequency bands (Fig. 4a–f). Furthermore, calculation of the Pearson correlation coefficient showed that, for the low theta, positive correlations between the 30 s versus 10 s curves were only significant after normalization of time, with the highest correlation being observed for coherence

(Supplementary Table 1). Altogether, these results indicate that BA-DMS interactions are involved in processing the CS–US time interval in aversive Pavlovian paradigms.

**Plasticity underlying learning of a new interval.** One interesting aspect of the SBF theory of timing is that integration and storage of various time intervals is proposed to occur through weighting

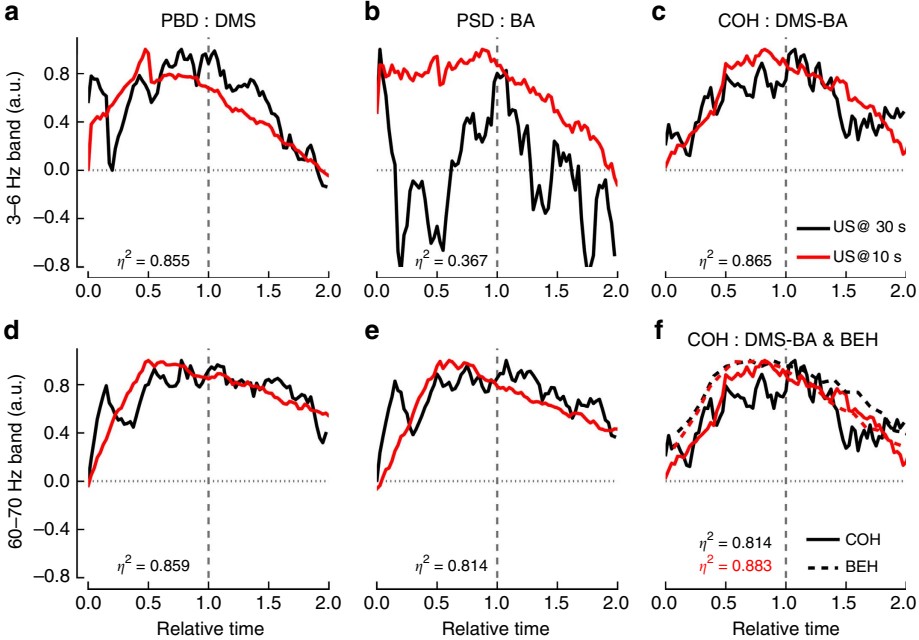

**Figure 4 | Superposition for coherence between DMS and BA LFP oscillations correlates with behaviour.** (**a–e**) The rescaled PSD in DMS (**a,d**), BA (**b,e**) and the coherence between DMS and BA (**c**) are shown for the 3–6 Hz (upper row, (**a–c**)) and the 60–70 Hz (**d–e**) frequency bands. After normalization of the time axis, the expected time of US arrival occurs at 1 (marked by the grey dashed line) for both the 30 s (black lines) and the 10 s conditions (red lines). $\eta^2$ values between the normalized 30 s and 10 s curves ($n = 81$ points) are given in each panel. (**f**) For illustrative purposes, superposition of the rescaled 3–6 Hz frequency band coherence traces (solid lines) with the normalized conditioned suppression curves (behaviour: BEH; dashed lines) for 30 s and 10 s conditions. The $\eta^2$ values correspond to the superposition index between coherence and behaviour for 30 s (black, $n = 60$ points) and 10 s (red, $n = 20$ points).

of cortico-striatal synapses[6]. As amygdalo-striatal afferents have been found to promote long-term plasticity of cortico-striatal pathways[28] we assessed whether such a plasticity mechanism could be detected following a change in the duration of the CS–US interval. We first performed immunostaining for the protein product of the immediate early gene Arc (activity-regulated cytoskeleton-associated protein), a marker of synaptic plasticity, in the DMS and dorsolateral (DLS) striatum in shifted (experimental) and non-shifted (control) rats (Fig. 5a,b). After 40 sessions of conditioned suppression training, animals were submitted to a single conditioning session with a 10 s CS–US interval (Fig. 5a). As expected, behavioural analysis revealed fast learning in the shifted group, as they showed a significant increase in suppression during the first 10 s before the US arrival during this shift session, compared with a baseline taken the day before the shift ($F_{1,16} = 4.90$, $P = 0.042$). No significant change in suppression was observed in the non-shifted control animals ($F_{1,20} = 3.84$, $P > 0.05$). Analysis of Arc expression in DMS and DLS at 4 time points (30, 60, 90 and 150 min) revealed an increase which followed the typical pattern of Arc protein expression, maximal 90 min after the beginning of the training session (Fig. 5c). This pattern of expression differed depending on group and brain area (significant group X brain area X perfusion time double interaction, $F_{3,36} = 5.80$, $P < 0.01$). Analysis at the 90 min time point showed that Arc expression was significantly higher in DMS, compared with DLS, for the control (non-shifted) animals ($F_{1,5} = 9.25$; $P = 0.03$; Fig. 5c, left panel). Most interestingly, in shifted animals Arc expression in the DMS was markedly reduced as compared to non-shifted animals (Fig. 5d, $F_{1,9} = 8.00$, $P = 0.02$); that is, increased to a lesser extent (Fig. 5c, right panel). Importantly, as there was no global difference in lever-pressing behaviour between these two 90 min subgroups of animals (neither in pre-CS mean lever-press per second $1.29 \pm 0.21$ versus $0.95 \pm 0.18$, nor during the 60 s

CS, mean suppression $0.53 \pm 0.02$ versus $0.56 \pm 0.03$, maximum suppression $0.87 \pm 0.07$ versus $0.88 \pm 0.07$; all Ps > 0.05), the difference in Arc labelling could not be related to an unspecific modification in general behaviour output, but rather to the detection of new temporal contingencies. Such a decrease in Arc up-regulation in the DMS suggests that plasticity mechanisms that are continuously taking place due to CS–US presentations are down-regulated as a result of replacement of the 30 s interval by a new duration.

Since reduction of Arc expression after the US shift implies a change in synaptic efficacy in the DMS, but not in the DLS, and since dense prefronto-DMS projections[29] are thought to play a critical role in temporal processing[7], we tested for plasticity at prefronto-DMS synapses induced by the shift of the CS–US interval. After an initial 40 sessions of conditioned suppression training, rats were implanted with stimulating and recording electrodes (Fig. 6a) and retrained for several sessions. We then recorded field potentials evoked in the DMS by prelimbic cortex (PL) electrical stimulation[30] each morning, while behavioural sessions continued in the afternoon (Fig. 6b). After recording a stable baseline of PL-DMS evoked field potentials (EFP) for four consecutive days, the arrival time of the US was shifted from 30 s to 10 s in a manner similar to the LFP oscillation and Arc experiments. As expected, this change in US arrival time induced a behavioural shift in the suppression curve toward the 10 s interval, from the first session onward (Fig. 6c). Further, this shift resulted in a marked decrease in synaptic efficacy in the PL-DMS pathway, as measured 24 h later (Fig. 6b; Day-1 versus Day 0, $F_{1,10} = 17.75$, $P < 0.01$), which remained stable over subsequent days (Day 0 to 3, $F < 1$). Together with the decrease in Arc expression in DMS following the shift in CS–US interval, the present LTD-like change in synaptic efficacy supports the SBF model which postulates that cortico-striatal plasticity subtends the learning and storage of time frames.

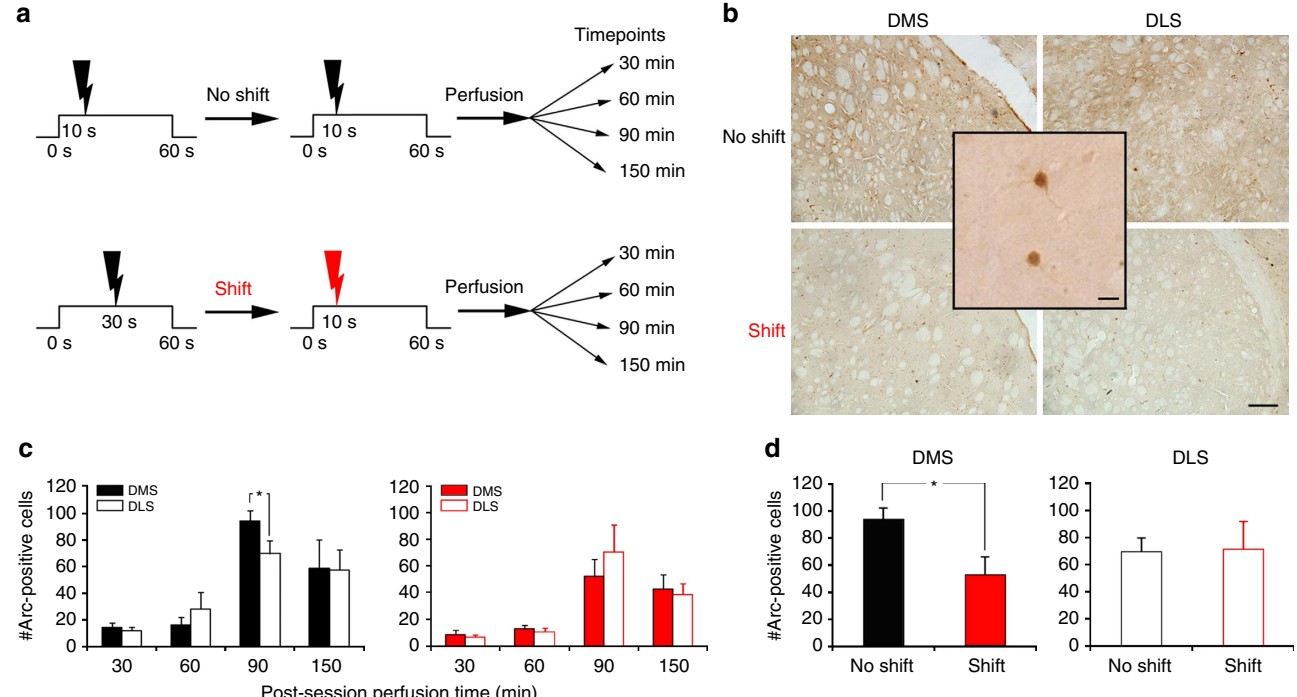

**Figure 5 | Arc expression is reduced in the DMS when the CS–US interval is shifted.** (**a**) Schematic explaining experimental design in test (shift) and control (no shift) groups. The 'shift' group underwent an aversive conditioning protocol with a CS–US interval of 30 s until the day of shift where such interval was shortened to 10 s. The control group was trained with a fixed CS–US interval of 10 s all along. With this design, the session animals underwent before perfusion was strictly the same for both groups. The only difference resides in the fact that the 10 s interval is a new duration for one group, and not for the other. Animals were perfused at different times after the last behavioural session (no-shift, $n = 6$ per time point; shift, $n = 5$ per time point). (**b**) Photomicrographs of transverse Arc-stained sections from representative cases illustrating expression in the DMS and DLS 90 min after the shifted versus non-shifted session. Scale bar, 200 μm. The inset shows a high-resolution image of DMS neurons immunostained for Arc. Scale bar, 10 μm. (**c**) Quantification of the number of striatal Arc-positive cells (mean + s.e.m.) for non-shifted (left panel) or shifted (right panel) animals perfused at different time points after the behavioural session revealed that Arc expression 90 min post-training was higher in the DMS than in the DLS for the non-shifted animals (ANOVA, *$P = 0.03$). (**d**) Quantification of the number of striatal Arc-positive cells (mean + s.e.m.) for DMS and DLS in non-shifted and shifted animals perfused at 90 min after the behavioural session shows a reduction in Arc expression in the DMS of shifted rats. ANOVA, *$P = 0.02$.

**Role of the amygdala in striatal plasticity and timing behaviour.** Next, we sought to specifically assess the role of the amygdala in the regulation of striatal plasticity in duration learning. For this, we first verified that BA is activated by the shift in CS–US interval by examining Arc staining detected 90 min after the 'shift' or 'no-shift' session (in the same animals as in Fig. 5). There was a significant increase in Arc immunostaining after the 'shift' session compared to 'no-shift' controls, indicating that the basolateral amygdala was indeed differentially activated following a change in CS–US duration (Fig. 7; $F_{1,9} = 9.48$, $P = 0.01$).

We then directly asked whether the amygdala is indeed a key regulator of DMS plasticity processes that occur during duration learning. To do so we asked whether the activation of the amygdala during the shift session controls the down-regulation of Arc-related plasticity mechanisms in the DMS. After training of the animals for more than 40 sessions of conditioned suppression, the animals were implanted bilaterally with *cannulae* aimed at the basolateral amygdala, and after recovery retrained for 11 sessions with a 30 s CS–US interval. Then, animals were given intra-amygdala infusion of either the sodium channel blocker lidocaine (shift lidocaine) or saline (shift saline) 10 min before a single session with a shift to a 10 s CS–US interval. Brains were subsequently harvested 90 min after the shift session and processed for Arc labelling (Fig. 8a). As in Fig. 5c (right panel), Arc expression was not different between DMS and DLS in the saline shift group ($F < 1$); in sharp contrast, level of Arc was

higher in the DMS than DLS in the group injected with lidocaine ($F_{1,4} = 21.56$, $P < 0.01$), as for non-shifted animals (Fig. 5c, left panel). As a result, the increased Arc expression was significantly higher in lidocaine infused animals with respect to saline controls specifically in the DMS (Fig. 8c, $F_{1,9} = 7.44$, $P = 0.02$ for DMS and $F < 1$ for DLS). Importantly, as neither the animal's reactivity to foot-shocks (difference in lever-pressing before and after the US delivery, $0.42 \pm 0.11$ versus $0.31 \pm 0.13$), nor the global lever-pressing activity ($0.83 \pm 0.18$ versus $0.87 \pm 0.18$) differed between the two groups ($Ps > 0.05$), the differences in Arc labelling were not related to global changes in animal's behaviour. Thus, in accordance with our hypothesis that the amygdala facilitates DMS plasticity induced by a change in CS–US interval, the decreased expression of Arc in the DMS was blocked by inactivation of the amygdala with lidocaine on the day of shift. These data support that the change in striatal plasticity processing occurring in DMS during learning of a new duration is under the control of the amygdala.

We also assessed the functional impact of amygdala inactivation on behavioural adaptation to the new temporal CS–US contingency. The protocol was identical to the previous experiment, except animals were infused during two shift sessions and their behaviour was followed during 10 additional drug-free training sessions with the new 10 s CS–US interval. A differential dynamic in learning the new CS–US interval between the lidocaine and saline groups was evidenced by a significant group X time X block interaction of suppression, when the analysis was

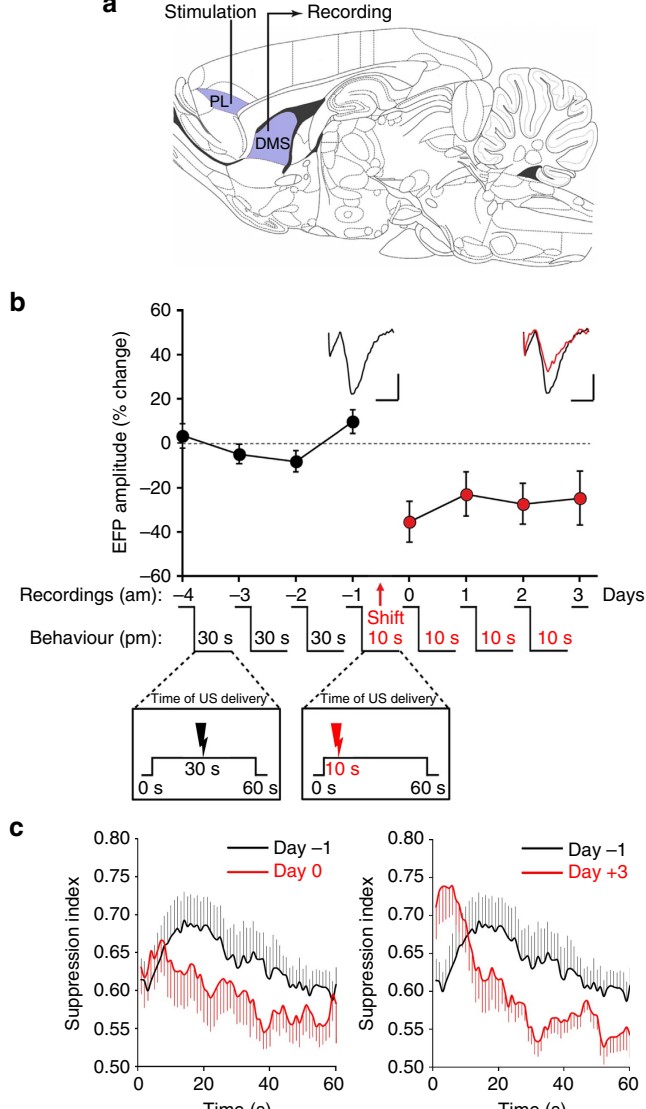

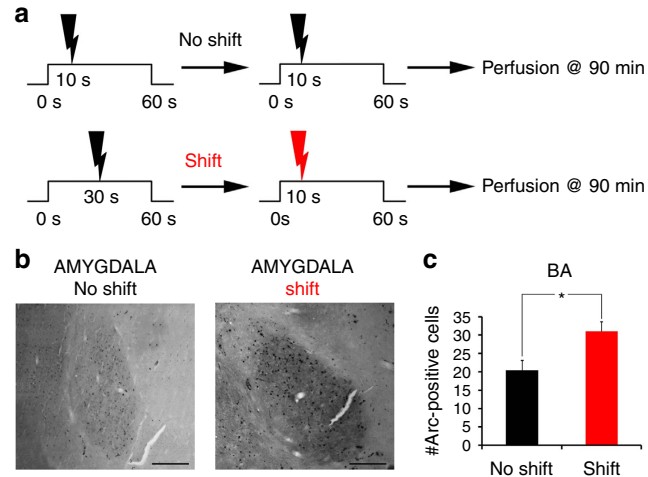

**Figure 6 | A shift in the CS–US time interval induces LTD-like long-term plasticity at prefronto-striatal synapses.** (**a**) Schematic diagram illustrating placement of the stimulating electrode in the prelimbic cortex (PL) and recording electrodes in the DMS, reproduced in part from ref. 64 with permission. (**b**) Mean ± s.e.m. changes in EFP amplitude for 4 days before (black circles) and 4 days after (red circles) a shift in the CS–US interval ($n = 11$). The experimental design is described on abscissa. EFP recording sessions were performed in the morning (am) and behavioural sessions in the afternoon (pm). Insets: example traces of the EFP before (black) and day + 3 after the shift (red). Scale bars, 2 mV, 10 ms. (**c**) Evolution of the conditioned suppression index ( ± s.e.m.) during the 60 s tone for the day before the shift (black lines) and the first (left panel) and last (right panel) day after the shift of the CS–US interval (red lines).

**Figure 7 | Amygdala is activated by shifting the CS–US interval.** (**a**) Schematic of explaining experimental design in test (shift) and control (no shift) groups. The 'shift' group underwent an aversive conditioning protocol with a CS–US interval of 30 s until the day of shift where such interval was shortened to 10 s. The control group was trained with a fixed CS–US interval of 10 s all along. Animals were perfused 90 min after the last behavioural session (no-shift, $n = 6$; shift, $n = 5$). (**b**) Arc immunostaining of the amygdala. Scale bar, 200 μm. (**c**) A shift in CS–US interval induced a significant increase in Arc staining (ANOVA, *$P = 0.01$), thereby showing that BA is activated as a result of the shift in duration. Slices were taken from the animals presented in Fig. 5d.

Adaptation of suppression behaviour to the new temporal contingency requires both expecting the US at a new (10 s) time interval as well as extinguishing the expectancy of the US at the old (30 s) time interval. While both processes may be reflected through a growing peak of suppression near the new time interval, the extinction of the old expectancy must be reflected through the shaping of the curve width. To characterize the impact of amygdala inactivation on either process, we thus further analysed these data by individually fitting Gaussian suppression curves for each rat in each session block and determined the evolution of the behavioural suppression peak time (index of duration learning) as well as the width (index of extinction processes) of the curve. Strikingly, this analysis revealed that inactivation of the amygdala with lidocaine did not delay learning of the new 10 s peak time (no group X block interaction within the first 2 blocks, $F_{1,17} = 1.36$, $P > 0.05$, Fig. 8e). Instead, the temporal pattern adapted faster for the lidocaine group, as the width was significantly narrower than for the saline group during the first block (significant group X block interaction, $F_{1,17} = 6.52$, $P = 0.02$; *post hoc* Bonferroni $P < 0.05$ for the first block, Fig. 8e). This result indicates that amygdala inactivation facilitates extinction of US expectation at the old 30 s duration. Insofar as the behavioural readout of the rat's temporal expectancy of the new CS–US interval is a function of both learning this new duration and extinguishing the old one, evolution of the suppression peak amplitude would thus, in fact, represent facilitation of the old duration extinction. To gain further insight on the role of the amygdala on extinction processes, we analysed the time at which rats started to suppress as well as the time at which they stopped on individual trials, as in Tallot *et al.*[19] In accordance with a facilitated extinction of the old interval, stop times appeared to reflect both learning of the new duration and extinction of the old one in the control group, whilst in the lidocaine group, extinction was already optimal early on after the shift (Fig. 9). Taking into account the blockade of

restricted to the first 10 s of the CS ($F_{16,272} = 1.82$, $P = 0.03$, Fig. 8d), while their pre-CS lever-pressing level remained stable (from $1.60 ± 0.16$ to $1.38 ± 0.13$ lever-press per second for lidocaine, and $1.39 ± 0.11$ to $1.23 ± 0.12$ for saline). The lidocaine group indeed showed a significant time X block interaction ($F_{16,128} = 2.77$, $P < 0.001$), whereas the control group did not ($F_{16,144} = 1.41$, $P > 0.05$), indicating a delay in stabilizing the new suppression behaviour at CS onset in amygdala inactivated animals.

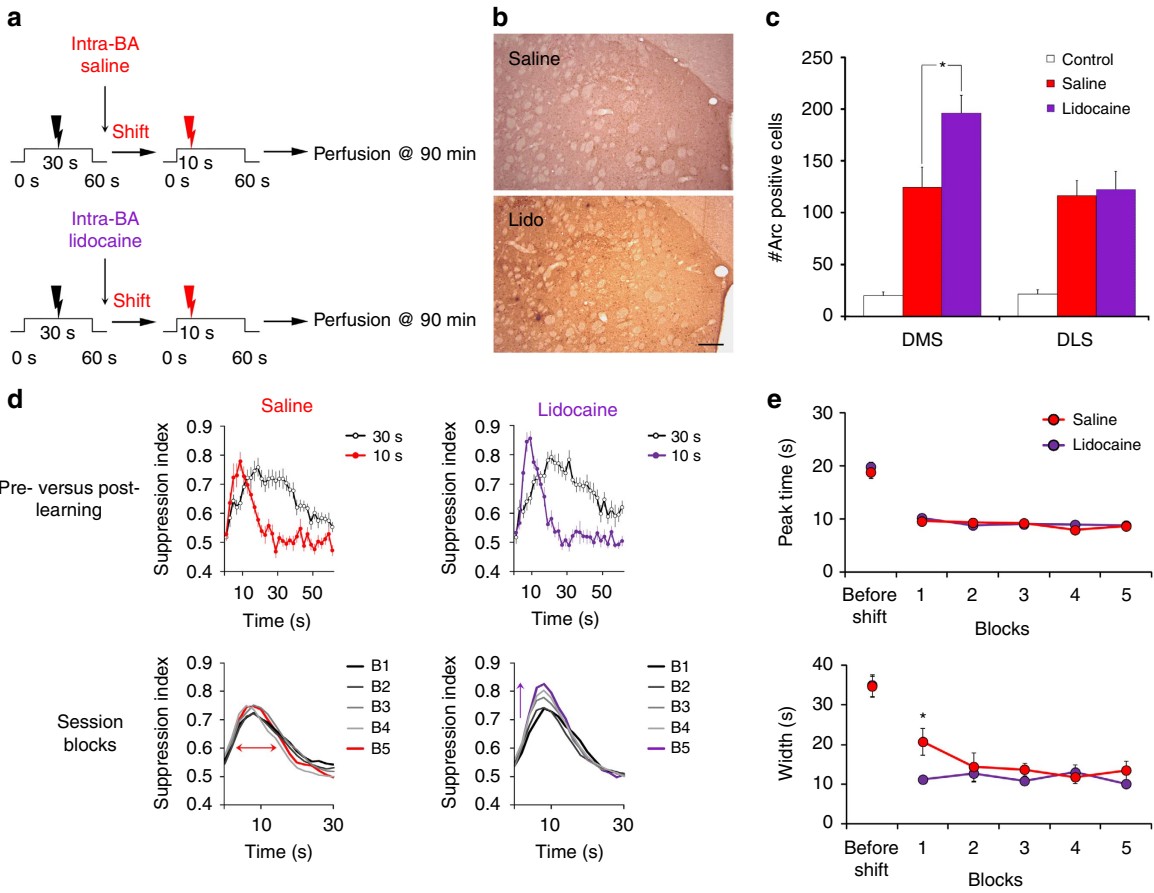

**Figure 8 | Amygdala inactivation facilitates extinction of duration memories.** (**a**) Schematic explaining experimental design in control saline and experimental lidocaine shift groups. Saline and lidocaine groups underwent an aversive conditioning protocol with a CS–US interval of 30 s until they were shifted to 10 s; saline or lidocaine was infused in the basolateral amygdala (20 μg μl$^{-1}$; 0.5 μl) just before the shift session. Rats were perfused and brains harvested 90 min after the shift session and processed for Arc immunohistochemistry (saline $n = 6$; lidocaine $n = 5$). (**b–c**) Immunostaining for Arc in DMS and DLS revealed that inactivating the amygdala prevents the reduction of Arc up-regulation in the DMS, but not in the DLS (ANOVA, $*P = 0.02$). Scale bar, 200 μm. These data indicate that blocking neuronal activity in the amygdala prevents the change in striatal plasticity processing occurring during learning of a new duration. Furthermore, control rats ($n = 5$) not exposed to the CS showed markedly low levels of Arc in both DMS and DLS. (**d**) For a behavioural assessment of the effect of amygdala blockade, additional rats were infused with saline ($n = 10$) or lidocaine ($n = 9$) before the first two shift sessions. Upper panels show behavioural patterns of suppression for both groups before the shift (US@30 s) and after 10 sessions of learning the new duration (US@10 s). In the lower panels analyses of the suppression curves by blocks of 2 sessions after the drug-infused sessions indicate that the lidocaine group continues to increase up to block 5, whilst in the control saline group no significant evolution of the peak amplitude could be detected. (**e**) Peakfit analyses revealed that acquisition of the peak time is very rapid as it occurred within the first block in both groups (upper panel). Strikingly, analysis of the width (red arrow in **d**), reflecting processes related to extinction of the old duration, shows non-immediate adaptation in controls whilst in the lidocaine group such process is immediate (*post hoc* Bonferroni, $*P < 0.05$).

shift-induced change in DMS plasticity by lidocaine inactivation of the amygdala, a conspicuous interpretation of these data would be that facilitation of PFC–DMS plasticity by the amygdala prevents the extinction of acquired durations and thereby helps to maintain duration memories.

## Discussion

This study was aimed at shedding light on the neural basis of interval timing. In particular, given the prominent involvement of the striatum and the emerging role of the amygdala in interval timing, we sought to determine whether both structures work in concert in the ability to estimate time intervals in the seconds to minutes range. Our dedicated design allowed us to study not only the rising expectancy during the CS period but also the decline in expectancy after the time of US arrival has passed, as well as whether the scalar property of timing holds at the neural level. Our analyses of LFP PSD and coherence confirm that both

structures are involved in temporal expectancy, as the power of both theta and gamma oscillations showed increases during the CS with a maximum that shifted as a result of a change in CS–US interval and its resulting behavioural peak time in US expectancy (from 30 s to 10 s). Most interestingly, a significant increase in coherence between these structures was found in the 3–6 Hz low theta band. Such data are reminiscent of the study by Popescu *et al.*[31] who, although not addressing the timing component of aversive conditioning, showed a strong coherence of neural oscillations in the gamma range between the basolateral amygdala and the posterior striatum in cats, which remarkably increased as the learning of an association between a tone CS and a reward US progressed; an explanation for the discrepancy between our two studies may lie in the recording site (anterior dorsomedial in our case and posterior ventrolateral in the previous study) within the striatum. When combined, the previous study and ours also raise the possibility that learning of the association *per se* is modulated by the amygdalo-striatal projection through gamma oscillations,

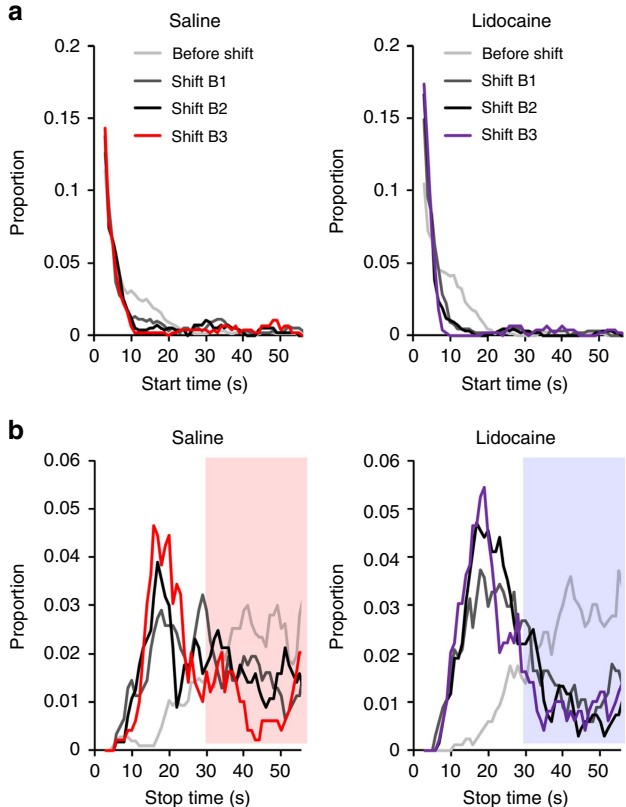

**Figure 9 | Intra-amygdala infusion of lidocaine facilitates extinction of responding associated to the old duration.** The figure shows the distribution of the times at which rats started to suppress after the CS onset ((**a**), start times) and at which they resumed back to their baseline lever-pressing ((**b**), stop times) when the CS–US interval was shifted from 30 s (before shift) to 10 s (for 3 blocks of 4 sessions). Stop times appeared to reflect both learning of the new duration (stops between 10 and 30 s) and extinction of the old one (stops >30 s, shaded area) in the saline group ($n = 10$), whilst in the lidocaine group ($n = 9$), extinction of responding to the old time was already optimal early on after the shift.

phase of hippocampal place cells with respect to the ongoing theta cycle has been found to advance as the spatial location of the animal evolves implying that sequences of place cell spiking are effectively compressed in time[38]. This phenomenon, known as phase precession, may underlie a theta/gamma discrete phase code which has been proposed to undertake more general brain coding schemes[39]. Interestingly, in place cells, phase precession was found to correlate with spatial but not temporal aspects of behaviour; the opposite might be true in medial spiny neurons of the DMS.

With regard to the SBF theory for interval timing, strengthening of specific patterns of cortico-striatal activation are thought to enable striatal memory storage of manifold intervals through reinforcement of cell/synaptic assemblies to be compared with ongoing patterns of cortico-striatal activations[6]. One interpretation of such theory is that multitudinous sets of cortico-striatal synapses are continuously being weighted across time intervals. In our configuration this would imply that the pattern of cortico-striatal synapses corresponding to the 30 s time interval was subjected to long-term changes in synaptic strength upon each presentation of the CS–US pairing. Such an interpretation is supported by our immunostaining of the immediate early gene and marker of synaptic plasticity Arc in the DMS and DLS, which showed a typical increase in protein expression 90 min after training[40,41] in animals that remained subjected to the same CS–US time interval, thus revealing a network undergoing recent synaptic changes. Strikingly, the LTD-like plasticity we observed as a result of a change in timing of the US arrival also supports this view. Indeed, if the 30 s interval is constantly being decoded and stored through synaptic reinforcement, a shift to the 10 s interval would cause the involved synapses to undergo depotentiation and settle at a different (reduced) synaptic strength. In accordance with this interpretation, Arc expression was less increased in the DMS (a region specificity previously implicated in several reports[21,22]) in animals in which reinforcement time was shifted to 10 seconds, suggesting that synaptic strengthening mechanisms were less prominent because of the shift in duration. Interestingly, whilst long-term potentiation would require sustained Arc expression[42,43], LTD has been associated with a transient reduction in Arc transcription, followed by an increase[43,44]. Although the precise functions of Arc in neural plasticity are complex, it is thought that different levels of Arc expression would allow for different synaptic changes by sliding the frequency threshold for strengthening or weakening of synaptic efficacy, as depicted in the BCM model of synaptic plasticity[45]. Such a view would be in agreement with our observation that the shift to a new CS–US interval is also associated with an up-regulation of Arc, but to a lesser extent than Arc expression triggered by the old duration. By this interpretation, a lower but still substantial amount of Arc expression would allow for a weakening of synaptic transmission. Thus, the LTD-like plasticity we observed might also reflect LTD mechanisms, rather than depotentiation, enabling behavioural adaptation to the new CS–US interval. It is worth mentioning a recent study showing that infusion of anisomycin, a protein synthesis inhibitor, into the dorsal striatum of rats did not prevent the rapid learning of a new time of reinforcement arrival in an appetitive instrumental peak interval paradigm[46]. Since both pre- and postsynaptic LTD co-exist at cortico-striatal synapses[47–51], one possibility is that learning of a new duration also involves presynaptic LTD mechanisms, which may be independent of protein synthesis[48]. Alternatively, such result could also support the view that depotentiation rather than bona fide LTD mechanisms are at play and do not involve gene expression. An important aspect to consider is the particular double-value the CS acquires in our

while the timing component of the CS–US association would be driven by theta oscillations. In accordance with this hypothesis, aversive conditioning in mice has been found to induce an increase in theta power in the amygdala, which progressively develops within the presentation of the CS as the time of US arrival approaches[32]. Similarly, in an appetitive instrumental task, a significant correlation between striatal theta rhythm and temporal behaviour in rats has recently been described[33]. The non-parametric cluster analysis used here shows that only specific sub-bands of the theta rhythm are timing-correlated and that coherent theta oscillations couple the striatum and amygdala during the timing of the CS–US interval. Moreover, our timing-directed designed experiments, which include testing for the scalar property, establish for the first time, direct evidence of amygdala-striatum coupling as neural correlates of interval timing.

Consistent with the facilitating role of the amygdala in associative memory[34,35] and cortico-striatal plasticity[28], frequencies in the theta range, alone or when combined with higher-frequencies, have been shown to promote long-term potentiation of cortico-striatal[36,37], including prefronto-striatal[30], synaptic efficacy. Such plasticity may rely in the temporal relationship of striatal spikes to the ongoing theta rhythm as found in hippocampal place cells[38]. Indeed, the firing

paradigm, that is, US predictor before the US time and safety (no-US) value after the US time, which results in the superposition of new learning for the 10 s duration and extinction for the old 30 s duration. Indeed, as both processes occur in parallel, the plasticity we observe at PFC–DMS synapses might be the result of either phenomenon, or both. Interestingly though, the fact that inactivating the amygdala blocks such plasticity whilst facilitating extinction, raises the possibility that this LTD-like plasticity actually underlies maintenance of the old time interval and competes with a continuous plastic process subserving learning of the new duration. Such interpretation is consistent with our previous observation that the reconsolidation of previously learned durations is amygdala-dependent[13]. If this view held true, one would predict similar changes regardless of the direction of the shift (long to short or short to long) in our paradigm, but not in a situation in which the US is co-terminating with the CS as it renders the two situations asymmetrical. It also remains possible that the interplay between blockade of plasticity and the facilitation of time interval updating involves another brain structure sharing connections with both the striatum and amygdala. In any case, the current set of data provides compelling evidence, at both the physiological and molecular levels, of the involvement of amygdala-dependent synaptic plasticity mechanisms in the DMS when learning new durations and extinguishing old ones, in agreement with the SBF, the foremost model in interval timing.

Our results showing immediate learning of the new duration but progressive extinction of the old one following the shift in CS–US interval support the view that the learning of specific time intervals occurs rapidly in aversive conditioning, despite the extensive amount of training required to enable behavioural expression of the temporal memory encoded with the CS–US association[12,24,52]. Indeed, it has been suggested that the temporal aspect of the learning experience is a pre-requisite for learning CS–US associations[5,53]. Our experimental design, whereby a new duration is imposed whilst all other components of the aversive conditioning have been acquired, reveals this fast acquisition of

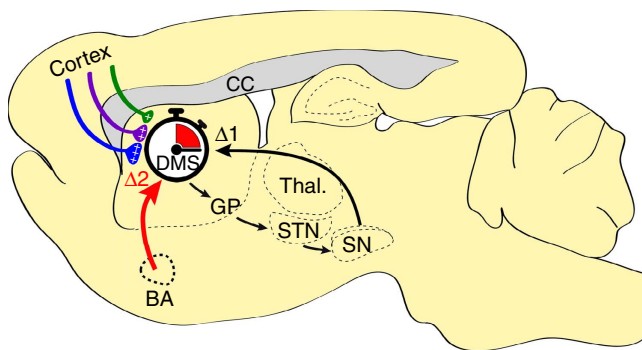

**Figure 10 | Proposed model of the role of the amygdala in the timing of CS–US interval.** The diagram depicts a set a cortical projections represented by coloured synaptic inputs connecting to the DMS, key brain region in interval timing. On encoding of a new duration, the cortico-striatal synapses would undergo different levels of synaptic strengthening represented by + signs. The DMS also receives dopaminergic inputs from the subtantia nigra pars compacta known to regulate such plasticity (Δ1) and thought to be involved in clock speed modulation. When detecting a change in temporal contingency, the amygdala would play a prominent role by settling cortico-striatal synaptic plasticity in a way that counteracts the synaptic strengthening induced by new duration learning through LTD and/ or depotentiation mechanisms (Δ2, red arrow), thus supporting the maintenance of already formed duration memories.

CS–US interval. Interestingly, however, this almost immediate learning of a new duration was not reported in appetitive conditioning timing protocols where an intermediate state was evidenced[46,54,55]. This difference might be attributable to the strong valence of the reinforcement when aversive, which will entail fast learning. Nevertheless, it raises the interesting question of whether the same neurophysiological and plasticity correlates would be detectable at the outset of learning, as our experiment targets the acquisition of a new duration, and its resulting behavioural adaptation, after overtraining. Notably, we previously observed changes in plasticity markers in the basolateral amygdala when shifting the CS–US interval after a single conditioning session[13], suggesting the involvement of a common network. However, it remains possible that only a subset of the correlates we observed here may underlie the flexible and fast adaptation to changes in temporal contingency. Further investigation would therefore be needed to decipher which networks and mechanisms subtend learning of interval, on the one hand, and flexible behaviour on the other.

The SBF model proposes a prominent role of dopaminergic projections from the Substantia Nigra pars compacta in the shaping of cortico-striatal synaptic inputs in relation to duration learning. Based on the large body of evidence showing that striatal plasticity is heavily regulated by dopamine[56,57] and that changes in dopaminergic signalling result in obvious distortions of time estimation, this influence indeed appears to be overt. However, the emerging role of the amygdala in interval timing[12,13], and the recently revealed control that it exerts on cortico-striatal plasticity[28], suggest that the latter structure is also a major player in interval timing, at least in aversive conditions. Our LFP coherence data taken together with the effect of inactivating the amygdala on striatal plasticity and temporal memories indeed suggest that this structure can also take part in interval timing of aversive events through direct control of cortico-striatal plasticity (Fig. 10). This conclusion is supported by previous investigations showing that the basolateral amygdala plays a role in the diverted attention processes engaged when timing in parallel an aversive and an appetitive cue, although without preventing timing per se[20,58]. The basolateral amygdala also plays a significant role in the updating of aversive memories when a change in CS–US interval is detected[12]. As a whole, these previous investigations and the current study indicate that the amygdala may not be required in the learning of new duration per se but nevertheless plays a prominent role in the regulation of the maintenance of already formed temporal memories.

In summary, using a behavioural protocol that enables specific assessment of temporal processes in an aversive conditioning paradigm, the current investigation provides neurophysiological and molecular evidence of the plasticity mechanisms proposed in the SBF to subtend estimation and learning of time intervals[6]. We propose that through the functional network involving the striatum and the amygdala, neural activity during routine conditions, or plasticity in the amygdala during the detection of a change in temporal contingencies, controls the cortico-striatal plasticity when learning a new duration, resulting in fast behavioural adaptation to the new contingencies. Finally, this amygdalo-striatal network provides a physiological basis for the influence of emotions on temporal estimation abilities[59].

## Methods

**Animals.** Experiments were carried out on 120 male Sprague-Dawley rats (Charles River Laboratories, France) in accordance with the guidelines of the European Community Council Directives of November 24th 1986 (86/609/EEC) and the French National Committee (87/848) for the care and use of laboratory animals. All efforts were made to minimize the number of animals used and their suffering. Rats were housed in standard laboratory cages and when necessary food restriction was performed as previously described[19]. Rats were randomly allocated to

experimental and control groups exept for the lidocaine experiment (Fig. 8) for which rat were assigned to aim for equivalent temporal behaviour between groups before the shift treatment. Data were analysed blind.

**Behaviour.** Behavioural training took place in a set of four identical conditioning chambers (30 × 25 × 30 cm, Coulbourn Instruments, USA), equipped with a lever, a food magazine connected to a pellet dispenser (45 mg pellets, Bioserv, USA), a shock floor and a speaker, all placed in a sound attenuating enclosure with a ventilation fan (65 db background noise). Behavioural protocols were controlled by Graphic State software (Coulbourn Instruments, USA).

Temporal precision of the conditioned response was assessed using a conditioned suppression paradigm[60], in which an aversive cue suppresses ongoing operant behaviour; that is, lever pressing for food on a variable interval schedule of 30 s on average (VI30). Aversive conditioning consisted of a 60 s tone CS (1 kHz, 80 dB) during which a mild electric footshock US (0.3–0.4 mA, 0.5 s) was delivered at a specific time after tone onset (30 s or 10 s). Importantly, the tone CS always lasted 60 s as it did not co-terminate with the US. In each experiment, reinforced CS–US and non-reinforced CS-alone (probe) trials were presented semi-randomly, allowing no more than 4 consecutive CS–US trials and no more than 2 consecutive probe trials. Only probe trials on which no shock US was presented were used to analyse the temporal pattern of suppression. A suppression ratio was calculated for each second of the 60 s CS from the mean number of lever presses per session according to the formula: 1—(mean number of presses in CS second $x$)/(mean number of presses in CS second $x$ + mean number of presses during inter-trial intervals). A ratio of 1 corresponds to complete suppression, 0.5 to no suppression. For analyses of peak time and width, each average curve was fitted using PeakFit software with a Gaussian function with a ramp as in Tallot et al.[19]

Behavioural training for analysis of amygdala-striatum functional connectivity consisted of 70 conditioning sessions with 8 CS–US trials and 4 CS alone trials followed by surgery, recovery, and re-training for 10 sessions before electrophysiological data acquisition. LFP recordings were performed on conditioning sessions consisting of 9 CS–US and 5 CS alone trials. Importantly, although animals were always connected to a recording cable, only alternating sessions in which the lever was removed from the chamber were considered for LFP analyses, to avoid electrophysiological activity due to motor control as opposed to activity related to timing processes. Thirty sessions were recorded before shifting the CS–US interval from 30 to 5 s.

For plasticity experiments involving Arc immunostaining, animals were trained for 40 sessions on a 30 s CS–US interval (experimental group) or 10 s CS–US interval (control group) schedule before shifting the experimental group to a 10 s CS–US interval schedule. Sessions consisting of 8 CS–US and 8 CS alone were alternated with sessions containing 8 CS–US only. The session of the shift happened on an 8 CS–US only day.

For amygdala inactivation experiments, animals were trained on a 30 s CS–US interval for 10 sessions with 8 CS–US trials, followed by 35 sessions with 8 CS–US and 8 CS alone trials, implanted with cannulae, and retrained under the 30 s CS–US interval schedule for 8 sessions. Then, animals chosen for the immunostaining assessment were exposed to sessions with only 8 CS–US trials two days before the shift day, followed by two sessions with 8 CS–US and 8 CS alone trials. On the shift day, animals were infused bilaterally with either lidocaine (20 µg µl[−1] diluted in saline), or saline at a rate of 0.5 µl/2.5 min per side (plus 1 min in place), and submitted 10 min later to a single session of shift to 10 s with 8 CS–US trials. Animals were then perfused 90 min later. Additional controls were perfused either after a VI30 lever-pressing session or directly taken from the colony room. Animals chosen for the behavioural assessment, were submitted to a second session of infusion/shifted CS–US trials, followed by 10 sessions with 8 CS–US with the shifted 10 s CS–US interval and 8 CS alone trials.

For plasticity experiments using EFP recordings, animals were trained for 40 sessions on a 30 s CS–US interval, followed by surgery, recovery and re-training for at least 8 sessions before shifting the CS–US interval to 10 s. Electrophysiological recordings (40 EFP recordings, 30 s inter-stimulation interval) were performed in another context in the morning, and animals continued to undergo conditioned suppression sessions in the afternoon. The day for shifting the CS–US interval was decided for each rat based on individual stability of EFP recordings over several days.

**Cannulae implantation.** Guide cannulae (PlasticOne, 26 gauge) were implanted under ketamine (75 mg kg[−1], i.p.) + domitor (50 µg kg[−1], i.p.) anaesthesia, with tolfedine (0.01 ml/100 g, i.p.) for analgesia, bilaterally into the basolateral amygdala (AP −3.0 mm, L ± 5.2 mm, DV −7.6 mm from skull). Once in place, they were fixed to the skull with dental acrylic cement, and dummy cannulae were inserted to prevent clotting. Surgery was followed by one week of recovery.

**Electrophysiology.** Recording electrodes were made from var-insulated nichrome wire (68 µm diameter). Wires were sharpened (0.7–1.0 MΩ) and placed in a 33 Gauge tube (PHYMEP, Paris, France), the tip extending 1 mm. Surgery was performed under pentobarbital anaesthesia (54.7 mg kg[−1], i.p.). Tolfedine (0.01 ml/100 g) and Robinul-V (0.01 mg kg[−1]) were given prior to surgery.

Concentric bipolar electrodes (300 µm tip separation) were used for stimulation. Surgery was followed by one week of recovery.

For LFP experiments, recording electrodes were implanted in the DMS (AP: 1.0 mm, L: 2.2 mm; DV: 3.2 mm ) and in the basal amygdala (AP −2.7 mm, L 4.7 mm, DV 8.5 mm). Reference and ground electrodes made of silver balls were placed epidurally over the cerebellum. Electrodes were assembled into a circular plug (Ginder Scientific, Canada, reference GS09PLG-220) and fixed on the skull with dental acrylic cement. During recording, LFPs were amplified 100x (Grass amplifiers, model P511), band-pass filtered (0.3 Hz-1 kHz) and acquired at 10 kHz in Spike2 via a CED interface (Power 1401 mkII, CED, UK). In total, the LFP was analysed from 8 rats before the shift (3 rats in 8 sessions with 5 CS-alone presentations; 5 rats in 9 sessions with 5 CS-alone presentations), and 6 rats after the US shift to 10 s (rats in 3, 5, 8, 9, 10 and 11 sessions with 5 CS-alone presentations).

For EFP experiments, the stimulating electrode was positioned in the prelimbic cortex (coordinates: AP 3 mm, L 0.8 mm and DV 2.5 mm) and the recording electrode in the ipsilateral DMS (AP 1 mm, L 2.2 mm and DV 3.2 mm). The depth of the recording electrode was adjusted to maximize the amplitude of the negative-going excitatory field potential. Daily recordings from 11 rats were performed in the morning and consisted of field potentials evoked by single stimulations (80–120 µs, 400 µA, adjusted to get a stable baseline) at a frequency of 0.033 Hz for 20 min. Signals were acquired through field effect transistors, amplified, band-pass filtered (0.1 Hz–3 kHz) and digitalized at 20 kHz using an ITC-16 computer interface analogue–digital (A/D) converter (Instrutech Corp.) coupled to a personal computer (MAC-G4, Apple Macintosh Inc.). Signals were collected using A/Dvance P3.61j software (Robert McKellar Douglas).

**LFP analysis to calculate PSD and coherence.** Raw LFP traces including a 60 s pre-CS period, the 60 s CS period and a 30 s post-CS period of each trial were used to calculate the PSD of single traces and the coherence (COH) between signals from striatum and amygdala.

The PSD and COH were computed based on 5 s windows centred on each time point, and time points were calculated every 0.25 s (that is, the 5 s analysis window was advanced in 0.25 s steps in the interval [−60, 90] s). The PSD was calculated using an adaptive weighted multitaper method. This was done by estimating the discrete prolate spheroidal sequences (orthogonal data windows), multiplying each of the tapers with the data series, compute the fast Fourier transform (FFT), and using the adaptive scheme for a better estimation of the discrete-spectrum weighted average[61]. The coherence between LFP signals was computed from the FFT and the weights of the multitaper spectrum estimation[62]. The analysis parameters of the multitaper method were as follows: time-bandwidth product = 3.5, number of used tapers = 7. The PSD and COH calculations are performed using mtspec which is a Python wrapper for the Multitaper Spectrum Estimation Library[62]. PSD and COH from all sessions (each session contained five CS-alone presentations, see total number of sessions in the 'Electrophysiology' section) before the shift in US arrival time and from session 6 onwards after the US shift were averaged, and are termed 30 s traces and 10 s traces, respectively. Artifacts, generated from movements of the animal during recordings, appeared as large deflections in the raw LFP traces. Epochs with artifacts were excluded from calculating the average PSD. To detect artifacts, we calculated the standard deviation (s.d.) of the raw signal in 5 s windows centred at each time point [−60, 90] s (step size 0.25 s). Time points for which the s.d. exceeded 1.3 times the median s.d. of all time points in a trial, as well as 1 s before and after, were excluded from the averaging. The s.d. threshold was determined upon visual inspection and based on the histogram of all s.d.'s in a trial, in which the artifacts appeared as clear outliers. An entire CS presentation was excluded if more than 60% of all time points were identified as artifacts. The averages for PSD and coherence pre- and post-shift were calculated for each rat by averaging across all trials and in all sessions taking into account the variable number of data points per time point due to the exclusion of artifacts.

Significant ($P < 0.05$) changes in the mean PSD and COH from baseline (based on the 60 s pre-CS period) were determined using a non-parametric cluster-level 1 sample $t$-test[63]. The procedure uses a cluster analysis with permutation test for calculating corrected $P$-values. Randomized data were generated with random sign flips, that is, the sign (+1 or −1) is flipped randomly for each data instance. If the data distribution on null hypothesis has zero mean, a random sign flip will not alter the mean. Thus permuting enables to test non-parametrically the null hypothesis. Non-parametric cluster tests were performed using the Python implementation of the MNE software package[63]. PSD and coherence are presented as z-scores with respect to the mean and the baseline per frequency band during the pre-stimulus interval ([−60, 0] s; Figs 2 and 3). For testing superposition, all curves (PSD, coherence and behaviour) are normalized to the maximal value during the stimulus period (Figs 1 and 4). The $\eta^2$ values are calculated based on those normalized curves.

All analyses routines were implemented in custom written Python scripts.

**Immunohistochemistry.** For Arc immunostaining, rats were perfused at 30, 60, 90 or 150 min after training. After a rapid deep anaesthesia with an overdose of pentobarbital, rats were transcardially perfused with phosphate-buffered saline (PBS) followed by 400 ml of ice-cold 4% paraformaldehyde (PFA) in 0.1 M phosphate buffer. Brains were removed, post-fixed overnight in 4% PFA and placed

in a cryoprotecting solution composed of 30% glycerol and 0.1% sodium azide in 0.1 M phosphate buffer. Free-floating sections (40 μm) containing the regions of interest were cut using a sliding microtome. Every sixth section was processed for Arc immunoreactivity. After blocking in PBS containing 1% bovine serum albumin-0.1% Triton X-100, slices were incubated overnight at room temperature in anti-Arc antibody (mouse monoclonal sc-17839, 1:500; Santa Cruz Biotechnology) in PBS containing 1% BSA-0.1% Triton X-100. After extensive washes in PBS, tissue sections were incubated with secondary antibody (Vectastain Anti-mouse IgG, biotynilated antibody 1:500; Elite PK-6102) in PBS-1% BSA. This was finally followed by washes and processing using the VectaStain Elite ABC kit (Vector Laboratories) and development in DAB peroxidase substrate for 5 min. Sections were mounted on electrostatic slides and coverslipped with DPX mounting medium. Images at × 10 or × 40 and were collected using an Olympus BX60 microscope (Leica Microsystemes, Germany) equipped with a CoolSNAP camera (Roper Scientific, USA) and Openlab software (Improvision, UK). Cell counting was performed in a defined region of interest (240 × 220 pixels) using Image J.

**Histology.** On completion of electrophysiological and pharmacological experiments rats were perfused with 4% paraformaldehyde. Brains were removed and post-fixed. Coronal sections 35 μm thick were then cut on a microtome and stained with thionin for identification of recording sites (see Supplementary Figs 3–5).

**Statistics.** Between groups comparisons ANOVAs were performed after verification of equal variance.

**Replication statement.** Behavioural interval timing experiments were replicated three times in total as they were necessary for LFP, EFP and lidocaine experiments.

**Data availability.** Data from the experiments presented in the current study are available from the corresponding author.

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

## Acknowledgements

We thank Gérard Dutrieux, Nathalie Samson-Desvignes, Pascale LeBlanc-Veyrac and Valérie Halin-Marsaux for technical help. We are also grateful towards Bruce L. Brown for helpful discussions. MG would like to thank Alexandre Gramfort for help with the non-parametric cluster analysis. This work was financed by ANR-07-NEURO-048–01 'Memotime', ANR-11-EMCO-012-02 'TDE', ANR-16-CE37-0004-04 'AutoTime' to V.D., NIH MH046516 and NIH MH038774 to J.E.L., CAPES 2350/09-2 and FAPESP grants 11/08575-7, 12/06825-9 to R.C.R.R.M. and FAPESP 2014/22178-9 to T.F.T. The collaboration between CNRS/Université Paris-Sud (VD lab) and New York University (JEL lab) was supported by CNRS-NYU LIA EmoTime and Partner University Funds 'Emotion & Time'.

## Author contributions

R.C.R.M., T.F.T. and J.K. contributed equally to this work. V.D., G.D., L.D.M. and G.E.S. designed the experiments; J.K., G.D., T.F.T., L.T., R.C.R.M., N.E.M., J.B.B., A.V. and S.H. performed the experiments; M.G., G.D., L.D.M., J.K., T.F.T., S.H., R.C.R.M., A.R. and V.D. analysed the data; M.G. contributed analysis tools; G.D., V.D., L.D.M., M.G. and J.E.L. wrote the paper.

## Additional information

**Competing financial interests:** The authors declare no competing financial interests.

