## [Peer Review File · Nature Communications]

Reviewers' comments:

Reviewer #1 (Remarks to the Author):

Review of Dallérac et al.

Overall, this paper addresses specific issues regarding the role(s) of the amygdala and striatum in interval timing. The questions addressed are highly novel/provocative and the experimental work is extremely well done. The findings hold great practical and theoretical importance for the field of timing and time perception. My recommendation would be for publication following revision along the lines outlined below.

In the abstract the behavioral procedure is referred to as "aversive conditioning", but in the text it is referred to as "threat (fear) conditioning". It would be best to be consistent. A "period" is missing at the end of the next to last sentence in the abstract.

The authors show that changes in coherence between striatum and amygdala LFPs were found to couple these structures during interval estimation within the 3-6 Hz theta rhythm. Why is the 3-6 Hz range referred to here as theta rhythm when other investigators typically refer to the 5-10 Hz or 7-14 Hz ranges as being theta rhythm with <5 Hz typically being in the delta range (e.g., DeCoteau et al., 2007)?

DeCoteau, W.E., Thorn, C., Gibson, D.J., Courtrmanche, R., Mitra, P., Kubota, Y., & Graybiel, A.M. (2007). Oscillations of local field potentials in the rat dorsal striatum during spontaneous and instructed behaviors. *Journal of Neurophysiology*, 97, 3800-3805.

They also show that a change only in CS-US interval results in long-term changes in cortico-striatal synaptic efficacy.

Meck & MacDonald (2007) investigated the role of the basolateral amygdala in the timing of aversive cues using a conditioned emotional response (CER) procedure similar to the employed in the current study. Moreover, this study demonstrated the superimposition of different CS durations (10 s and 20 s) paired with electric shock.

In the current study, the maximum response suppression supported by the threatening CSs occurred at earlier time points than the target durations. This suggests that rats are beginning to return to the baseline level of responding supported by the VI 30-s schedule of food reinforcement once they pass the temporal criterion sampled from the 30-s (or 10-s) distribution rather than using an upper threshold for responding that would normally extend beyond this criterion by a proportional amount of time (see Meck & MacDonald, 2007).

Meck, W.H., & MacDonald, C.J. (2007). Amygdala inactivation reverses fear's ability to impair divided attention and make time stand still. *Behavioral Neuroscience*, 121, 707-720.

The inhibition of protein synthesis in the dorsal striatum during the acquisition of a new temporal criterion didn't affect the transition from the initial state (baseline A) to the intermediate state, but it did affect the transition from the intermediate state to the final state (baseline B) suggesting a differential dependency of these two processes on protein synthesis (MacDonald et al., 2012). In the current study, the existence of an intermediate state isn't so obvious. This may be due to more rapid learning in fear conditioning preparations or the direction of the shift (e.g., lower to higher duration in MacDonald et al. (2012) and higher to lower duration in the current study). See Lejeune et al., 1997, Meck et al. (1984), and Rodriguez-Girones & Kacelnik (1999) for additional details of this intermediate transition phase.

Lejeune, H., Gerrara, A., Simons, F., & Wearden, J.H. (1997). Adjusting to changes in the time of

reinforcement: Peak-interval transitions in rats. *Journal of Experimental Psychology: Animal Behavior Processes*, 23, 211-231.

MacDonald, C.J., Cheng, R.K., & Meck, W.H. (2012). Acquisition of "Start" and "Stop" response thresholds in peak-interval timing is differentially sensitive to protein synthesis inhibition in the dorsal and ventral striatum. *Frontiers in Integrative Neuroscience*, 6:10.

Meck, W.H., Komeily-Zadeh, F.N., & Church, R.M. (1984). Two-step acquisition: Modification of an internal clock's criterion. *Journal of Experimental Psychology: Animal Behavior Processes*, 10, 297-306.

Rodríguez-Gironés, M.A., & Kacelnik, A. (1999). Behavioral adjustment to modification in the temporal parameters of the environment. *Behavioural Processes*, 45, 173-191.

The authors report measures of cortico-striatal plasticity and also LFP data between the striatum and amygdala. A more complete dataset would include fronto-striatal LFP data and a measure of plasticity between the striatum and amygdala. The former seems appropriate given that the article is addressing the striatal beat-frequency model. A summary of the simulation parameters for the striatal beat-frequency (SBF) theory of interval timing is provided in the Appendix of Allman & Meck (2012). This is useful in terms of evaluating the electrophysiological properties of the model and comparing them with the findings in the current study.

Allman, M.J., & Meck, W.H. (2012). Pathophysiological distortions in time perception and timed performance. *Brain*, 135, 656-677.

Plasticity experiments: In the experimental group, the duration was shifted from 30 to 10 sec. This will cause both new learning for the 10-sec duration and extinction for the old 30-sec duration. Consequently, the LTD/plasticity change in the experimental group could be due to either of these processes, but the authors seem to attribute the result to the new learning at 10-sec entirely. A second control group that is trained at 30-sec, and is simply extinguished (i.e., no shift, just extinction of the 30-sec duration) might remedy this problem. If the LTD differences are absent in this control, then one would be able to claim that the LTD observed in the experimental group is due to learning the new 10-sec duration. Furthermore, another group could be used that learns a 10-sec duration, but doesn't extinguish at the old 30-sec duration. For example, shocks could be delivered at either of these durations on different trials. LTD here would be reflective of learning the new 10-sec duration alone. The authors suggest the amygdalostriatal projection may mediate some of the plasticity effects seen in the data. Inactivation of this circuit with DREADDs and including the above mentioned groups, in addition to the ones already included, would provide better evidence for this (i.e., inactivating the circuit and finding a lack of plasticity effects).

LFP experiments: The coherence between the amygdala and striatum is an important part of this report. However, an inactivation technique would also be helpful for this point in order to determine whether the observed synchronization could be due to the involvement of some other brain area that keeps time and shares connections with both of these structures. For example, the PFC sends projections to the striatum and to the amygdala, which has been implicated in fear conditioning. If the PFC keeps time, this activity might cause both the striatum and amygdala to synchronize with it during a trial. This would make it appear as if the striatum and amygdala are synchronizing with each other, even though this would be coincidental to their synchronization with the PFC.

Is the amygdala-striatal connection only involved in the timing of aversive events, i.e., how general are these findings? I would recommend that the authors point out that several previous studies have lesioned or inactivated the amygdala using the peak-interval procedure and have found virtually no effects on timing performance (e.g., Olton et al., 1987). The way the article is currently written makes it seem as if the authors have demonstrated that the amygdala is

involved in all forms of interval timing.

Reviewer #2 (Remarks to the Author):

Dallérac et al. show that a shift in interval timing (between a CS and a shock US) causes a corresponding shift in the time at which theta frequency in the 3-6 Hz range in the dorsomedial striatum is maximal, and in the time at which coherence in this LFP frequency range between lateral amygdala and DMS is maximal. The scalar property of timing (variance proportional to mean) was evident in the distribution of theta spectral density across the established and shifted intervals. These results are consistent with a role for LA-DMS interactions in the time component of CS-US associations. In addition, the authors show reduced numbers of Arc-positive neurons in the DMS after the temporal shift, as well as smaller evoked potentials in DMS from prefrontal cortical stimulation. These results are interpreted as evidence of a change in corticostriatal synaptic strength that may contribute to the shift in interval timing.

This paper has several very positive features, including the use of probe trials in a timing task (allowing the scalar property of variables related to timing to be assessed), the observation of neural activity related to a shift in timing, and the evidence for LA-DMS interactions in timing. These results are novel and intriguing. However, these interesting findings do not relate in any obvious way to the plasticity findings; rather, the plasticity observations test a different hypothesis. The end result is that neither hypothesis is tested as thoroughly as it could be. Specifically, all of the evidence is correlative, with no interventional experiments to demonstrate necessity of any of the observed phenomena for timing or for learning a shift in timing.

In addition, there are a few interpretational problems with the plasticity experiments. For instance, the authors observe a decrease in Arc in the DMS after shifting the CS-US interval from 30 to 10 sec. As the authors propose, this decrease could be related to synaptic plasticity, but they have not ruled out an obvious alternative hypothesis: the animal's behavior is different in the 10 vs 30 sec conditions; the reduced Arc expression could be related solely to differences in behaviors that result from the difference in timing, not in the timing mechanism itself. The EFP experiment mitigates this concern, but only somewhat. That experiment has its own problems: could it be that the lower magnitude of evoked potentials after the shift is due to some global change in behavior state and/or in neuromodulator or neurohormonal levels (dopamine, norepinephrine, serotonin...)? Perhaps tighter temporal coupling of the CS and US results in such changes, which might then cause differences in neuronal excitability that are independent of (or a product of) the timing mechanism.

In sum, the paper would be strengthened by showing that the variables that they show to be correlated with shifts in timing actually represent processes that cause those shifts.

Reviewer #3 (Remarks to the Author):

In this paper the authors show that during the expectancy of an aversive stimulus upon a conditioned tone, the dorsal striatum and the amygdala exhibit an increase in LFP coherence in the 3-6 Hz band, which the authors refer to as theta. These structures also display increased power in the 3-6 Hz band, although the change in power was only statistical for the striatal recordings. Very interestingly, the time in which the peak of the 3-6 Hz coherence occurs shifts when the time in which the aversive stimulus is (or would be) delivered is changed. The authors also show in this paper that the number of Arc positive cells is reduced in the dorsomedial striatum (DMS) of animals subjected to the CS-US time shift protocol compared to a control group, and also that

these animals have lowered evoked response in the DMS to electrical stimulation of the prelimbic cortex.

I find this paper interesting but quite intriguing at the same time. Below I list some issues that called my attention.

I missed some sort of control group in which the same tone would not be associated with shock; what would happen to the LFPs in this case? Also, I know things are easier said than done, but ideally the authors could also have an experimental group in which the US is delivered first at 10 s, and then would shift to 30 s. Or else, further ideal would be to have multiple time shifts of different magnitude (and not only a single one) to convincingly demonstrate that the changes in the time course of coherence levels really follow the time shift.

A natural question is whether the results could be explained by behavioral/locomotor differences between both protocols (10 s and 30 s). Are the authors confident that this was not the case? Was for instance locomotor activity measured and compared?

The authors should show some measure of dispersion (SD) or confidence (SEM, CI) for their behavioral metric (i.e., the suppression index). Also, a statistical analysis for the shift in peak time of the suppression index shown in Figure 1B would be welcome.

The authors should inform how they normalized time in Figure 1C ("relative time").

Why in Figure 1B the peak of the red curve is higher than the peak of the black curve, while in Figure 1C both curves have the same height? Wasn't the normalization performed only in the time axis? Confusing...

I missed some sort of statistics for proving that the results really follow the scalar property (by the way, the authors may want to define somewhere in the paper what is meant by the scalar property). Computing a single eta squared for the average curves does not seem very convincing to me. Ideally one should test for this scalar property on an individual subject basis... what about performing some sort of correlation or cross-correlation within animals? Or else, perhaps the authors could show the (paired) distribution of the width of the curves as well as normalized peak times? For instance, the authors wrote "As predicted by the scalar property, the width of the curve was reduced proportionally to the time shift, as there was good superposition of both curves", but there is no stats to back this claim. (And related to my first point above, the authors write "proportionally to the time shift", but only a single value of time shift was investigated.... Ideally this proportionality should be assessed using multiple time shift magnitudes...).

The authors could show some examples of non-normalized PSDs. As a reader, I would be interested to know whether the power spectrum exhibits any power peak. And similarly for the coherence spectrum.

I would also be interested to know whether the reported effects are specific for theta and gamma frequency ranges; since the authors show spectra for 0 to ~15 Hz and ~55 to ~95 Hz, one naturally wonders what is happening to the other frequencies.

The authors measured arc expression 30, 60, 90 and 150 minutes after the behavioral protocol in "shift" animals. However, the authors only report results for the 90 min group. What about the other time points? In short, the authors should display the same bar graphs as in Figure 5C (which was done for the control animals) for the "shift" animals as well.

Related to the point above, the writing gives the impression of a reduced arc expression in "shift" animals, but in reality what is (likely) happening is that arc expression is actually increased, but to

a lesser extent than in controls.

In their text, the authors try to relate the reduced arc expression to LTD; is there any evidence for this? A quick search in the literature (and admittedly non-exhaustive) revealed that arc expression may actually be required for LTD (Waung et al, Neuron 2008; reference 36 cited by the authors also show a transient increase)...

The authors state "As down-regulation of Arc expression after the US shift implies a change in synaptic efficacy in the DMS, but not in the DLS, (...) we tested for plasticity at prefronto-DMS synapses induced by the shift of the CS-US interval." Ideally, if the authors want to prove that the change in arc expression is associated to a specific change in synaptic efficacy in DMS but not DLS, why not making the natural control of also recording evoked responses in DLS?

There is a parenthesis missing in the formula of the suppression index (pg13-14).

In the methods, the authors wrote: "Importantly, LFPs were also recorded in alternating sessions in which the lever was removed from the chamber, in order to be able to dissociate electrophysiological activity due to motor control from electrophysiological activity related to timing processes." ... I got confused with the statement of "were also recorded" and caught myself wondering which sessions were actually used in the analysis shown in the paper... was there a difference between recordings in the presence vs absence of the lever?

For computing coherence, one needs multiple time windows to average FFT vectors. What was the time window used in this study? I understand that the coherogram was obtained using 5-s sliding windows, but for computing the coherence spectrum of each 5-s window, this should be further divided into smaller windows...

Could the authors better explain the permutation test used to check the stats of the cluster analysis? In particular, what random sign flips means?

I find a bit odd that the closing sentence of the abstract focuses on the amygdala, while several of the main findings (changes in power, evoked response, Arc+ cells) were obtained for the striatum. Also, there is a punctuation mark (.) missing before this sentence.

In Figure 4, the abbreviation BEH is not introduced.

In Figure 2F bottom, there is a " $p < 0.1$ " which I believe may be a typo.

Reviewer #4 (Remarks to the Author):

The manuscript by Dellerac et al studies coherence between striatum and amygdala during aversive conditioning, with particular emphasis on the learning that occurs when the intervals between the CS and US are shifted. The key findings are that changes in the coherence (local field potentials) between amygdala and striatum were correlated with interval changes, and that such changes also correlated with changes in corticostriatal plasticity.

The manuscript is an interesting one, it is novel, it addresses a neglected question, and it reports an enormous amount of time consuming work. I congratulate the authors on their persistence.

Upon my reading the data were largely handled appropriately (but see below re suppression ratio and preCS rates), appropriate credit is assigned to previous work, and the text is clear and well written.

I had the following comments on the experiments and interpretation.

1. The key manipulation here is shifting the delivery of the shock reinforcer, because this reveals the presence of timing at the behavioral level and its neural correlates in amygdala and striatum. I did not understand why the authors only shifted backwards - from 30s CS - US interval to a 10 s interval. The theories they are testing predict that the same changes should occur in animals shifted from 10s to 30s. The point here is that shifting backwards causes a net reduction in overall suppression and presumably fear, and hence it is unclear whether the changes in neural synchronization are due to changes in timing or simply to reductions in fear. If these changes in synchronization were due to learning the new intervals, as the authors and models predict, then they should be observed in animals shifted from 10s - 30s. It may be that the authors did this and I missed it.; if not, it would certainly merit close attention in a discussion.

2. The authors present 'timing' as fundamental to aversive learning, and of course they are probably correct. But, they used a very large number of training sessions to reveal such evidence for timing. This is not problematic - in fact it is typical of the timing literature. However, it does raise the question of the primacy and functional significance of timing in the aversive system. That is, animals express evidence for fear/aversive/threat conditioning many trials before they express evidence for timing of the conditioned response. I think the authors could do a better job of addressing this. It does not reduce the significance of what they have reported, but it is theoretically and functionally important to address (how many prey species have this much training with predation? Is this timing a 'secondary' system fine tuning defensive behavior?).

3. The ARC data are interesting, but suffer from the absence of a control condition. We have ARC data from shifted and non-shifted rats and the groups differ. This difference is interesting and important, but it is difficult to interpret. Which group is up and which is down?

4. My final substantive comment is that it would be very useful to see some kind of causal manipulation of these circuits on the timing behaviours to determine if they really are critical to the timing. In the absence of such causal data, we are looking at correlations, albeit very interesting ones.

These substantive points deserve some consideration in the manuscript. These are long and time consuming experiments - the manuscript is interesting and novel as it stands. It may be that addressing most of these issues in the text will be sufficient.

Minor points

1. The authors use 'fear', 'threat', and 'aversive' interchangeably. I would prefer to see one term (ideally aversive or fear) used throughout. It is confusing for readers not expert in the area, or who do not understand the provenance of these terms, to see these different terms.
2. The suppression ratios were calculated in an unusual way. The standard way of calculating a suppression ratio is $CS/(CS - preCS)$. This is the approach that has been used for nearly 50 years. Does the way these ratios are calculated influence the evidence?
3. Finally, it is important to report preCS lever pressing rates as suppression ratios are heavily dependent on these.

Response to referees

'The referees, who have expertise in neural circuitry of reward and fear, interval timing, and spectral analysis techniques, are requesting (1) additional control experiments to rule out alternative explanations for the observed effects and (2) manipulation experiments that would more conclusively demonstrate the involvement of the amygdala and striatum.

Reviewer #1 (Remarks to the Author):

Overall, this paper addresses specific issues regarding the role(s) of the amygdala and striatum in interval timing. The questions addressed are highly novel/provocative and the experimental work is extremely well done. The findings hold great practical and theoretical importance for the field of timing and time perception. My recommendation would be for publication following revision along the lines outlined below.

1. In the abstract the behavioral procedure is referred to as "aversive conditioning", but in the text it is referred to as "threat (fear) conditioning". It would be best to be consistent. A "period" is missing at the end of the next to last sentence in the abstract.

We agree with the reviewer and now refer to the conditioning paradigm we used as "aversive conditioning" throughout the text.

The period has been added.

2. The authors show that changes in coherence between striatum and amygdala LFPs were found to couple these structures during interval estimation within the 3-6 Hz theta rhythm. Why is the 3-6 Hz range referred to here as theta rhythm when other investigators typically refer to the 5-10 Hz or 7-14 Hz ranges as being theta rhythm with <5 Hz typically being in the delta range (e.g., DeCoteau et al., 2007).

*DeCoteau, W.E., Thorn, C., Gibson, D.J., Courtemanche, R., Mitra, P., Kubota, Y., & Graybiel, A.M. (2007). Oscillations of local field potentials in the rat dorsal striatum during spontaneous and instructed behaviors. *Journal of Neurophysiology*, 97, 3800-3805.*

Although the reviewer's question is indeed reasonable, we feel uncomfortable referring to 3-6 Hz range as delta rhythm since the latter covers frequencies that can go much lower than 3 Hz. Furthermore, several investigations actually refer to similar 3-6 Hz range as "low theta rhythm"; e.g. Eckart et al. 2014 (2-4 Hz), Salisbury and Taylor 2012 (3-5 Hz), Cervera-Ferri et al., 2011 (3-6 Hz). Therefore, we now refer to the 3-6 Hz band as low theta rhythm and the 6-9 Hz as high theta rhythm (similarly to Cervera-Ferri et al., 2011; Eckart et al., 2014).

*Cervera-Ferri, A. et al. Theta synchronization between the hippocampus and the nucleus incertus in urethane-anesthetized rats. *Exp. Brain Res.* 211, 177–192 (2011).*

*Eckart, C., Fuentemilla, L., Bauch, E. & Bunzeck, N. Dopaminergic stimulation facilitates working memory and differentially affects prefrontal low theta oscillations. *Neuroimage* 94, 185–92 (2014).*

Salisbury, D. F. & Taylor, G. *Semantic priming increases left hemisphere theta power and intertrial phase synchrony. Psychophysiology* 49, 305–11 (2012).

3. They also show that a change only in CS-US interval results in long-term changes in cortico-striatal synaptic efficacy.

Meck & MacDonald (2007) investigated the role of the basolateral amygdala in the timing of aversive cues using a conditioned emotional response (CER) procedure similar to the employed in the current study. Moreover, this study demonstrated the superimposition of different CS durations (10 s and 20 s) paired with electric shock. In the current study, the maximum response suppression supported by the threatening CSs occurred at earlier time points than the target durations.

Meck, W.H., & MacDonald, C.J. (2007). Amygdala inactivation reverses fear's ability to impair divided attention and make time stand still. *Behavioral Neuroscience*, 121, 707-720.

We thank the reviewer for pointing out this study. We indeed confirm, using a within subject design, the anticipatory nature of the timing of the peak suppression response that Meck and McDonald previously reported, and that we also confirmed in a thorough behavioral analysis in our recent publication (Tallot et al., 2016). We now refer to this aspect of both papers in our revised manuscript p.5, l.122-124:

"The maximum conditioned suppression was at a time close to the US arrival, although anticipatory (average peak time at 22.5 ± 0.9 s; Fig. 1B), confirming previous reports using similar procedures^{19,20}."

Tallot, L., Capela, D., Brown, B. L. & Doyère, V. Individual trial analysis evidences clock and non-clock based conditioned suppression behaviors in rats. *Behav. Processes* 124, 97–107 (2016).

The inhibition of protein synthesis in the dorsal striatum during the acquisition of a new temporal criterion didn't affect the transition from the initial state (baseline A) to the intermediate state, but it did affect the transition from the intermediate state to the final state (baseline B) suggesting a differential dependency of these two processes on protein synthesis (MacDonald et al., 2012). In the current study, the existence of an intermediate state isn't so obvious. This may be due to more rapid learning in fear conditioning preparations or the direction of the shift (e.g., lower to higher duration in MacDonald et al. (2012) and higher to lower duration in the current study). See Lejeune et al., 1997, Meck et al. (1984), and Rodriguez-Girones & Kacelnik (1999) for additional details of this intermediate transition phase.

Insofar as our investigation analyzes the shift from one duration to another, the reviewer's suggestion to analyze the transition phase is indeed judicious. However, the dataset reported in Figure 1 does not enable us to analyze such a transition phase with sufficient precision because too many sessions were devoted to electrophysiological recordings (lever removed from the chamber). Still, the new dataset we report Figure 7E (upper panel) indicates that learning of the new peak time is indeed very rapid as it occurs within the first session block (i.e., after only 16 CS-US trials), suggesting no, or discrete, intermediate state. As pointed out by the reviewer, this suggests that learning a new duration is more rapid in aversive conditioning. Because extinction of the old duration

could take longer to achieve in aversive conditioning situations, as suggested by the non-immediate change in width following the shift in CS-US duration (Figure 7E lower panel) and the start-stop analysis we now report in supplementary Figure 4, this may also be influenced by the direction of the temporal change, although this was not found to modify learning kinetics of a new duration in an appetitive protocol (Meck et al., 1984). This issue has now been mentioned in the discussion of the revised manuscript p.17, l.441-445:

"Interestingly though, such almost immediate learning of a new duration was not reported in appetitive conditioning timing protocols where an intermediate state was evidenced^{45,53,54}. This difference might be attributable to the strong valence of the reinforcement when aversive, which will entail fast learning."

Meck, W. H., Komeily-Zadeh, F. N. & Church, R. M. Two-step acquisition: modification of an internal clock's criterion. J. Exp. Psychol. Anim. Behav. Process. 10, 297–306 (1984).

4. The authors report measures of cortico-striatal plasticity and also LFP data between the striatum and amygdala. A more complete dataset would include fronto-striatal LFP data and a measure of plasticity between the striatum and amygdala. The former seems appropriate given that the article is addressing the striatal beat-frequency model.

The main hypothesis of our study regards a role of the amygdala in influencing the cortico-striatal plasticity provoked by duration learning. Although LFP analysis of the fronto-striatal pathway would be interesting to investigate during interval timing, it does not focus on testing the latter hypothesis and should therefore be addressed in future studies. Similarly, while we hypothesize a role of amygdalo-striatal network, we do not specifically hypothesize that BA-STR **plasticity** mechanisms could be involved in the formation of temporal memories, as higher activity of the pathway may in itself suffice to modulate the cortico-striatal plasticity (Popescu et al. 2007). Thus, we feel that performing a measure of a putative behaviorally induced amygdalo-striatal plasticity would not bring further light to our leading hypothesis that activation of the amygdala influences prefronto-striatal plasticity. Given that the amygdalo-striatal EFP response has never been characterized, performing this experiment would be hazardous and time consuming, whilst not directly assessing our main working hypothesis. Hence, we instead provide an Arc staining of the amygdala (Supplementary Figure 3) indicating increased levels 90min after the shift and thereby showing that BA is indeed activated as a result of the shift in duration and thus supporting our hypothesis. Furthermore, we also report that amygdala inactivation at the time of the shift blocks the shift-related Arc modulation observed specifically in the dorsomedial striatum (Figure 7C), further strengthening our hypothesis of a functional role of the amygdalo-striatal network. This is explained in the relevant result section of the revised manuscript, p. 10-11, l. 262-293:

"Next, we sought to specifically assess the role of the amygdala in the regulation of striatal plasticity in duration learning. For this, we first verified that BA is activated by the shift in CS-US interval by examining Arc staining detected 90 min after the "shift" or "no-shift" session (in the same animals as in Fig. 5). There was a significant increase in Arc immunostaining after the "shift" session compared to "no-shift" controls, indicating that the basolateral amygdala was indeed

differentially activated following a change in CS-US duration (Supplementary Fig. 3; $F_{1,9} = 9.48$, $p = 0.01$).

We then directly asked whether the amygdala is indeed a key regulator of DMS plasticity processes that occur during supra-second duration learning. To do so we asked whether the activation of the amygdala during the shift session controls the down-regulation of Arc-related plasticity mechanisms in the DMS. After training of the animals for more than 40 sessions of conditioned suppression, the animals were implanted bilaterally with cannulae aimed at the basolateral amygdala, and after recovery retrained for 11 sessions with a 30 s CS-US interval. Then, animals were given intra-amygdala infusion of either the sodium channel blocker lidocaine (shift lidocaine) or saline (shift saline) 10 min before a single session with a shift to a 10 s CS-US interval. Brain were subsequently harvested 90 min after the shift session and processed for Arc labelling (Fig. 7A). As in Fig. 5C (right panel), Arc expression was not different between DMS and DLS in the saline shift group ($F < 1$); in sharp contrast, level of Arc was higher in the DMS than DLS in the group injected with lidocaine ($F_{1,4} = 21.56$, $P < 0.01$), as for non-shifted animals (Fig. 5C left panel). As a result, the increased Arc expression was significantly higher in lidocaine infused animals with respect to saline controls specifically in the DMS (Fig. 7C, $F_{1,9} = 7.44$, $P = 0.02$ for DMS and $F < 1$ for DLS). Importantly, as neither the animal's reactivity to foot-shocks (difference in lever-pressing before and after the US delivery, 0.42 ± 0.11 vs. 0.31 ± 0.13), nor the global lever-pressing activity (0.83 ± 0.18 vs. 0.87 ± 0.18) differed between the two groups ($P_s > 0.05$), the differences in Arc labelling were not related to global changes in animal's behavior. Thus, in accordance with our hypothesis that the amygdala facilitates DMS plasticity induced by a change in CS-US interval, the decreased expression of Arc in the DMS was blocked by inactivation of the amygdala with lidocaine on the day of shift. These data support that the change in striatal plasticity processing occurring in DMS during learning of a new duration is under the control of the amygdala."

Popescu, A. T., Saghyan, A. A. & Paré, D. NMDA-dependent facilitation of corticostriatal plasticity by the amygdala. Proc. Natl. Acad. Sci. U. S. A. 104, 341–6 (2007).

5. A summary of the simulation parameters for the striatal beat-frequency (SBF) theory of interval timing is provided in the Appendix of Aypellman & Meck (2012). This is useful in terms of evaluating the electrophysiological properties of the model and comparing them with the findings in the current study.

Allman, M.J., & Meck, W.H. (2012). Pathophysiological distortions in time perception and timed performance. Brain, 135, 656-677.

Our data do support the SBF model by showing an implication of oscillatory activity (although not in the hypothesized 10 Hz, but in 3-6Hz range) and plasticity (LTD-like) in the striatum in relation to interval timing, and we do refer to it when appropriate in the discussion (see p.14-16, l.379-432). As far as we know, the SBF model did not make any assumption on a potential role of the amygdala and/or amygdalo-striatal coherent activity. We thus feel that a more elaborate discussion of the SBF would be beyond the scope of the present manuscript.

6. *Plasticity experiments: In the experimental group, the duration was shifted from 30 to 10 sec. This will cause both new learning for the 10-sec duration and extinction for the old 30-sec duration. Consequently, the LTD/plasticity change in the experimental group could be due to either of these processes, but the authors seem to attribute the result to the new learning at 10-sec entirely. A second control group that is trained at 30-sec, and is simply extinguished (i.e., no shift, just extinction of the 30-sec duration) might remedy this problem. If the LTD differences are absent in this control, then one would be able to claim that the LTD observed in the experimental group is due to learning the new 10-sec duration. Furthermore, another group could be used that learns a 10-sec duration, but doesn't extinguish at the old 30-sec duration. For example, shocks could be delivered at either of these durations on different trials. LTD here would be reflective of learning the new 10-sec duration alone.*

We acknowledge that, in our protocol, learning the new duration is indeed concomitant with extinction of the old time interval. The experiments suggested by the reviewer are conceptually accurate and would indeed provide insights that may allow deciphering whether PFC-DMS plasticity is actually attributable to learning the new time interval or extinction of the old, or both. The current set of experiments sought to assess whether updating a time interval is associated with plasticity of cortico-striatal transmission, as predicted by the SBF theory, and did not aim at deciphering which of the processes is responsible for such changes. Nevertheless, the new dataset we provide in Figure 7 indicates that shutting down amygdala activity during the shift session blocks the aforementioned plasticity and facilitates extinction of the old 30 s duration, thus strongly suggesting that the LTD we observe as a result of the duration shift would, as indeed suggested by the reviewer, be at play at least in extinction processes. Our experiments therefore suggest that several mechanisms underlying learning a new duration and extinguishing the old are concomitantly at play. In-depth analysis of these intricate processes will be worth pursuing in future dedicated studies.

We have now implemented the manuscript with these results p.10-11, l.262-293 (see response to reviewer 1 point #4) and p.11-13, l. 294-336:

" We also assessed the functional impact of amygdala inactivation on behavioral adaptation to the new temporal CS-US contingency. The protocol was identical to the previous experiment, except animals were infused during two shift sessions and their behavior was followed during 10 additional drug-free training sessions with the new 10 s CS-US interval. A differential dynamic in learning the new CS-US interval between the lidocaine and saline groups was evidenced by a significant group X time X block interaction of suppression, when the analysis was restricted to the first 10 s of the CS ($F_{16, 272}=1.82$, $P=0.03$, Fig. 7D), while their pre-CS lever-pressing level remained stable (from 1.60 ± 0.16 to 1.38 ± 0.13 lever-press per second for Lidocaine, and 1.39 ± 0.11 to 1.23 ± 0.12 for saline). The lidocaine group indeed showed a significant time X block interaction ($F_{16,128}=2.77$, $P<0.001$), whereas the control group did not ($F_{16,144}=1.41$, $P>0.05$), indicating a delay in stabilizing the new suppression behavior at CS onset in amygdala inactivated animals.

Adaptation of suppression behavior to the new temporal contingency requires both expecting the US at a new (10 s) time interval as well as extinguishing the expectancy of the US at the old (30 s) time interval. While both processes may be reflected through a growing peak of suppression near the new time interval, the extinction of the old expectancy must be reflected through the shaping of the curve width. In order to characterize the impact of amygdala inactivation on either process,

we thus further analyzed these data by individually fitting Gaussian suppression curves for each rat in each session block and determined the evolution of the behavioral suppression peak time (index of duration learning) as well as the width (index of extinction processes) of the curve. Strikingly, this analysis revealed that inactivation of the amygdala with lidocaine did not delay learning of the new 10 s peak time (no group X block interaction within the first 2 blocks, $F_{1,17}=1.36$, $P>0.05$, Fig. 7E). Instead, the temporal pattern adapted faster for the lidocaine group, as the width was significantly narrower than for the saline group during the first block (significant group X block interaction, $F_{1,17}=6.52$, $P=0.02$; post-hoc Bonferroni $P<0.05$ for the first block, Fig. 7E). This result indicates that amygdala inactivation facilitates extinction of US expectation at the old 30 s duration. Insofar as the behavioral readout of the rat's temporal expectancy of the new CS-US interval is a function of both learning this new duration and extinguishing the old one, evolution of the suppression peak amplitude would thus, in fact, represent facilitation of the old duration extinction. In order to gain further insight on the role of the amygdala on extinction processes, we analyzed the time at which rats started to suppress as well as the time at which they stopped on individual trials, as in Tallot et al. (2016)²⁰. In accordance with a facilitated extinction of the old interval, stop times appeared to reflect both learning of the new duration and extinction of the old one in the control group, whilst in the lidocaine group, extinction was already optimal early on after the shift (Supplementary Fig. 4). Taking into account the blockade of shift-induced change in DMS plasticity by lidocaine inactivation of the amygdala, a conspicuous interpretation of these data would be that facilitation of PFC-DMS plasticity by the amygdala prevents the extinction of acquired durations and thereby helps to maintain duration memories."

7. The authors suggest the amygdalostratial projection may mediate some of the plasticity effects seen in the data. Inactivation of this circuit with DREADDs and including the above mentioned groups, in addition to the ones already included, would provide better evidence for this (i.e., inactivating the circuit and finding a lack of plasticity effects).

We agree with the reviewer that an interventional experiment would provide more compelling data to support our hypothesis. We have therefore performed additional experimentation whereby we inactivated the amygdala during shift sessions and performed an immunostaining for Arc in the DMS and DLS in order to assess striatal plasticity processes. For this we used a broad neuronal inhibitor, the sodium channels blocker lidocaine, with the aim of inactivating as much neuronal activity as possible in the BA. The use of DREADDs or Muscimol would have implied cell specificity and therefore a weaker blockade than lidocaine.

In accordance with our hypothesis that the amygdala facilitates DMS plasticity induced by a change in CS-US interval, the diminution of Arc activation we report in Figure 5D and that we replicate here in Figure 7C (no difference in Arc staining between DMS and DLS), is blocked by inactivation of the amygdala with lidocaine the day of the shift. These data indicate that blocking the amygdala prevents the change in striatal plasticity processing occurring during learning of a new duration and/or extinguishing the old duration.

We have now implemented the manuscript with these new interventional data and their relevance in the result section p. 10-11, l. 262-293 (see response to reviewer 1 point #4).

8. LFP experiments: The coherence between the amygdala and striatum is an important part of this report. However, an inactivation technique would also be helpful for this point in order to determine whether the observed synchronization could be due to the involvement of some other brain area that keeps time and shares connections with both of these structures. For example, the PFC sends projections to the striatum and to the amygdala, which has been implicated in fear conditioning. If the PFC keeps time, this activity might cause both the striatum and amygdala to synchronize with it during a trial. This would make it appear as if the striatum and amygdala are synchronizing with each other, even though this would be coincidental to their synchronization with the PFC.

This is an interesting suggestion. However, many structures actually share connections with both striatum and amygdala, including several cortices such as indeed the prefrontal cortex but also somatosensory and primary sensory areas; other brain regions projecting to both structures also comprise the *substantia nigra*, the thalamus, the hypothalamus or the hippocampus. Thus, we need to stress that testing all these structures to find out whether some other brain area influence both striatum and the amygdala is beyond the objective of the current investigation. We here sought to investigate whether duration learning induces prefronto-striatal plasticity and whether this process is influenced by the amygdala. The fact that the coherence between the striatum and amygdala increases during temporal stimuli corroborates our results indicating that activity of the amygdala indeed influences prefronto-striatal neurotransmission/plasticity. Whether this coherence is inherent to the amygdala driving striatal activity or whether it results from the activity of another brain region is a different and challenging question that should be addressed in a future investigation.

9. Is the amygdala-striatal connection only involved in the timing of aversive events, i.e., how general are these findings? I would recommend that the authors point out that several previous studies have lesioned or inactivated the amygdala using the peak-interval procedure and have found virtually no effects on timing performance (e.g., Olton et al., 1987). The way the article is currently written makes it seem as if the authors have demonstrated that the amygdala is involved in all forms of interval timing.

We agree with the reviewer that our data do not inform us on whether the amygdala is involved in the same way in all forms of interval timing, and the literature suggests that its critical role may be restricted to aversive conditions. However, it remains possible that the amygdala would not be required in the learning of new duration *per se*, whether aversive or appetitive, but nevertheless plays a prominent role in the regulation of the maintenance of already formed temporal memories. As recommended by the reviewer we have now modified the discussion in our revised manuscript in order to place our work in perspective with the few studies intervening on amygdala whilst using different protocols p.17-18, l.462-468:

"Our LFP coherence data taken together with the effect of inactivating the amygdala on striatal plasticity and temporal memories indeed suggest that this structure can also take part in interval timing of aversive events through direct control of cortico-striatal plasticity. Such conclusions are supported by previous investigations showing that the basolateral amygdala plays a role in the diverted attention processes engaged when timing in parallel an aversive and an appetitive cue, although without preventing timing *per se*^{19,57}. "

Reviewer #2 (Remarks to the Author):

Dallérac et al. show that a shift in interval timing (between a CS and a shock US) causes a corresponding shift in the time at which theta frequency in the 3-6 Hz range in the dorsomedial striatum is maximal, and in the time at which coherence in this LFP frequency range between lateral amygdala and DMS is maximal. The scalar property of timing (variance proportional to mean) was evident in the distribution of theta spectral density across the established and shifted intervals. These results are consistent with a role for LA-DMS interactions in the time component of CS-US associations. In addition, the authors show reduced numbers of Arc-positive neurons in the DMS after the temporal shift, as well as smaller evoked potentials in DMS from prefrontal cortical stimulation. These results are interpreted as evidence of a change in corticostriatal synaptic strength that may contribute to the shift in interval timing.

This paper has several very positive features, including the use of probe trials in a timing task (allowing the scalar property of variables related to timing to be assessed), the observation of neural activity related to a shift in timing, and the evidence for LA-DMS interactions in timing. These results are novel and intriguing. However, these interesting findings do not relate in any obvious way to the plasticity findings; rather, the plasticity observations test a different hypothesis. The end result is that neither hypothesis is tested as thoroughly as it could be. Specifically, all of the evidence is correlative, with no interventional experiments to demonstrate necessity of any of the observed phenomena for timing or for learning a shift in timing.

As explained in response to reviewer 1's comment #6 & #7, we agree that an interventional experiment would provide more compelling data to support our hypothesis. We have therefore performed additional experimentation whereby we inactivated the amygdala during shift sessions, using the sodium channels blocker lidocaine, and performed an immunostaining for Arc in the DMS and DLS in order to assess striatal plasticity processes. In accordance with our hypothesis that the amygdala facilitates DMS plasticity induced by a change in CS-US interval, the diminution of Arc activation we report in Figure 5D and that we replicate here in Figure 7C, is blocked by inactivation of the amygdala with lidocaine the day of the shift. These data indicate that blocking the amygdala prevents the change in striatal plasticity processing occurring during learning of a new duration.

To further assess the role of the amygdala on striatal plasticity and duration learning, we have also performed an additional behavioral experiment where activity of the amygdala is inhibited using lidocaine during 2 initial sessions of shift in order to assess whether such inactivation results in a delay in the behavioral adaptation to the new CS-US contingency. Analysis of the evolution of the suppression curve upon the 5 first 2-session blocks shows a significant time X group X blocks interaction when the analysis is restricted to the first 10 s, i.e. the new CS-US interval. The group receiving lidocaine indeed shows a significant interaction time X blocks while the control group does not, indicating a delay in stabilizing the new suppression behavior at CS onset in amygdala inactivated animals. We further analyzed these data by individually fitting suppression curves of each rat for each session block and, using a peakfit analysis, determined the evolution of the behavioral peak time as well as the width of the curve. Surprisingly, such analysis revealed that inactivation of the amygdala with lidocaine does not delay learning of the new 10 s peak time. However, through analyses of the curve's width and start-stop suppression behavior (Tallot et al., 2016), we unraveled

differences between the saline vs. lidocaine group indicating that processes underlying extinction of the 30 s target are compromised by inhibition of the amygdala. Inactivation of the amygdala during the initial error detection phase therefore appears to facilitate extinction of the old 30 s duration. Insofar as the behavioral readout of the rat's temporal expectancy of the new CS-US interval is a function of both learning this new duration and extinguishing the old one, evolution of the suppression peak amplitude would thus, in fact, represent facilitation of the old duration extinction. Taking into account the blockade of shift-induced change in DMS plasticity by lidocaine inhibition of the amygdala, a conspicuous interpretation of these data would be that facilitation of PFC-DMS plasticity by the amygdala prevents the extinction of acquired durations and thereby helps to maintain duration memories.

As a whole, these interventional data indicate that the amygdala influences striatal plasticity and thereby tunes duration learning and memory. Thus, the present set of data also provides mechanistic insights on how emotions could influence time estimation.

We have now implemented the manuscript with these new interventional data and their relevance in the result section p.10-13, l.261-336.

Talbot, L., Capela, D., Brown, B. L. & Doyère, V. Individual trial analysis evidences clock and non-clock based conditioned suppression behaviors in rats. Behav. Processes 124, 97–107 (2016).

1. In addition, there are a few interpretational problems with the plasticity experiments. For instance, the authors observe a decrease in Arc in the DMS after shifting the CS-US interval from 30 to 10 sec. As the authors propose, this decrease could be related to synaptic plasticity, but they have not ruled out an obvious alternative hypothesis: the animal's behavior is different in the 10 vs 30 sec conditions; the reduced Arc expression could be related solely to differences in behaviors that result from the difference in timing, not in the timing mechanism itself.

This is a reasonable question. In order to control for global behavioral differences between the shifted and non-shifted groups for which a difference in Arc expression was found, we have quantified the number of lever presses during the pre-CS (i.e. ITI) and during the CS, as well as the average suppression during the CS. There was no significant difference in lever-pressing behavior, nor in suppression behavior (mean or maximum), indicating that the difference in Arc expression in the DMS was not reflecting a global difference in the behavioral output, but rather the detection of a change in CS-US interval.

Furthermore, we have performed additional experiments in which the shift-related reduced Arc expression was blocked by infusion of lidocaine in the amygdala (Figure 7A-C), while again the global behavior (i.e. reactivity to footshocks, average lever-pressing) was comparable between both saline and lidocaine groups. These data therefore clearly show that the change in Arc expression is not attributable to a simple change in behavior, but rather to the detection of a change in CS-US temporal contingency.

Finally, as the reviewer points out (point #2 below), the EFP experiment showing a long-lasting change in field potentials observed 24h after the shift while there was no global change in lever-

pressing the day before, renders unlikely an interpretation of Arc modifications in terms of global difference in behaviors.

We have now implemented the manuscript with these information, p.9, l.235-240:

"Importantly, as there was no global difference in lever-pressing behavior between these two 90-min subgroups of animals (neither in pre-CS mean lever-press per second 1.29 ± 0.21 vs. 0.95 ± 0.18 , nor during the 60 s CS, mean suppression 0.53 ± 0.02 vs. 0.56 ± 0.03 , maximum suppression 0.87 ± 0.07 vs. 0.88 ± 0.07 ; all $P_s > 0.05$), the difference in Arc labelling could not be related to an unspecific modification in general behavior output, but rather to the detection of new temporal contingencies."

2. The EFP experiment mitigates this concern, but only somewhat. That experiment has its own problems: could it be that the lower magnitude of evoked potentials after the shift is due to some global change in behavior state and/or in neuromodulator or neurohormonal levels (dopamine, norepinephrine, serotonin...)? Perhaps tighter temporal coupling of the CS and US results in such changes, which might then cause differences in neuronal excitability that are independent of (or a product of) the timing mechanism.

For assessing neurotransmission efficacy at prefronto-DMS synapses we recorded evoked field potential at the same time of day every morning whereas behavioral training was performed in the afternoon, i.e. several hours later (Figure 6B). Thus, the change in EFPs was recorded ~20h after the temporal CS-US shift. Therefore, the global change in behavior state the reviewer suggests is an unlikely option as it would have to last for ~20h to underlie the change in EFP we report here. Indeed, variations in neuromodulators levels proposed to underlie such global change are typically associated to a relatively short time-scale (Lee and Dan, 2012) incompatible with the period of ~1 day between last behavioral session and EFP recording in our protocol.

Lee, S.-H. & Dan, Y. Neuromodulation of brain states. Neuron 76, 209–22 (2012).

Reviewer #3 (Remarks to the Author):

In this paper the authors show that during the expectancy of an aversive stimulus upon a conditioned tone, the dorsal striatum and the amygdala exhibit an increase in LFP coherence in the 3-6 Hz band, which the authors refer to as theta. These structures also display increased power in the 3-6 Hz band, although the change in power was only statistical for the striatal recordings. Very interestingly, the time in which the peak of the 3-6 Hz coherence occurs shifts when the time in which the aversive stimulus is (or would be) delivered is changed. The authors also show in this paper that the number of Arc positive cells is reduced in the dorsomedial striatum (DMS) of animals subjected to the CS-US time shift protocol compared to a control group, and also that these animals have lowered evoked response in the DMS to electrical stimulation of the prelimbic cortex.

I find this paper interesting but quite intriguing at the same time. Below I list some issues that called my attention.

1. I missed some sort of control group in which the same tone would not be associated with shock; what would happen to the LFPs in this case?

The control suggested by the reviewer would indeed be informative on the baseline reactivity of the amygdalo-striatal network to a 60 s tone. We now report recordings performed in rats submitted to the CS only (1kHz tone, no US) for two consecutive days. Analyses of DMS and BA PSD and of the coherence between these structures revealed no changes relating to the time passing by during the tone. We have now added these data as a Supplementary Figure 2 and mention it in the revised manuscript p.7-8, l.188-193:

"In order to control for the effect of the 60 s tone itself on LFP, we analyzed theta and gamma bands in both DMS and BA during 2 consecutive days of CS exposure without US delivery in naive rats, and found no significant variation in PSD and coherence (Supplementary Fig.2), thus confirming that the variation in LFP oscillations in conditioned animals were attributable to expectation of US arrival."

2. Also, I know things are easier said than done, but ideally the authors could also have an experimental group in which the US is delivered first at 10 s, and then would shift to 30 s. Or else, further ideal would be to have multiple time shifts of different magnitude (and not only a single one) to convincingly demonstrate that the changes in the time course of coherence levels really follow the time shift.

Although the reviewer's suggestion is interesting, we must stress that testing the direction of the shift or multiple time shifts is not a pre-requisite to assess our hypothesis that the amygdala influences the adaptation of temporal expectancy by controlling striatal plasticity. Besides, previous investigations on appetitive conditioning have found no bias on behavioral or physiological responses with regard to the direction of the shift in time interval (Meck et al., 1984; Mello et al., 2015). Furthermore, to be meaningful, a new dataset including multiple time shifts or a 10 s to 30 s shift would have to encompass all levels of analysis provided in the manuscript, that is, 1/ Behavioral analysis and assessment of the scalar property, 2/ LFP oscillation analyses during behavior, 3/ Arc immunostaining after the shift in an independent experiment, 4/ EFP recording before and after the shift in another independent behavioral experiment, 5/ analysis of the behavioral consequences of inactivating the amygdala and 6/ analysis of the effect of inactivating the amygdala on striatal plasticity. Given the extensive time required to train rats to our timing dedicated auditory aversive conditioning experimental paradigm and the time-consuming analyses involved, we think that doubling the dataset with a different shift in CS-US time interval would be unreasonable. This will, however, be worth pursuing in later studies as a direct follow-up of our present findings.

Meck, W. H., Komeily-Zadeh, F. N. & Church, R. M. Two-step acquisition: modification of an internal clock's criterion. J. Exp. Psychol. Anim. Behav. Process. 10, 297–306 (1984).

Mello, G. B. M., Soares, S. & Paton, J. J. A scalable population code for time in the striatum. Curr. Biol. 25, 1113–22 (2015).

3. A natural question is whether the results could be explained by behavioral/locomotor differences between both protocols (10 s and 30 s). Are the authors confident that this was not the case? Was for instance locomotor activity measured and compared?

As explained in the response to points #1 & #2 of reviewer 2, the plasticity results (Arc and EFP) cannot be explained by global changes in behaviors, as the average lever-pressing behavior both between trials and during the CS did not differ between the two conditions and changes in plasticity were observed the days after the temporal shift. As for the LFP correlates, they show a temporal pattern related to the US expectancy. We did not record the behavior during the electrophysiological recording sessions, as freezing is not a sensitive measure of timing within this time range in adult rats (Diaz-Mataix et al., 2013; Shionoya et al., 2013). The LFP correlates may coincide with animal's preparation to the expected US, possibly indexed by other behaviors, such as respiration or USVs (Shionoya et al., 2013). Regardless of the behavior, this would result from the US expectancy, and it would be difficult to disentangle in which direction the causal relationship is, as they would both be correlates of temporal expectancy.

Díaz-Mataix, L., Ruiz Martinez, R. C., Schafe, G. E., LeDoux, J. E. & Doyère, V. Detection of a temporal error triggers reconsolidation of amygdala-dependent memories. Curr. Biol. 23, 467–72 (2013).

Shionoya, K. et al. It's time to fear! Interval timing in odor fear conditioning in rats. Front. Behav. Neurosci. 7, 128 (2013).

4. The authors should show some measure of dispersion (SD) or confidence (SEM, CI) for their behavioral metric (i.e., the suppression index).

SEM have now been added to Figures 1 and 6 suppression index curves.

5. Also, a statistical analysis for the shift in peak time of the suppression index shown in Figure 1B would be welcome.

We have now indicated the statistical details concerning the shift in peak time of the suppression index in the relevant result section p.5, l.124-127:

"shifting the CS-US interval from 30 s to 10 s yielded an immediate shift in the peak of suppression (before shift vs. 1st session of shift: time X session interaction, $F_{59,295}=1.94$, $p<0.001$, Fig.1B) leveling off at a proportional reduction in peak time (8.6 ± 0.7 s) within 5 sessions"

6. The authors should inform how they normalized time in Figure 1C ("relative time").

Time was normalized to the effective time of occurrence of the US. In other words, time was divided by 30 for the US@30s data and by 10 for the US@10s data. We have now specified this in the legend of Figure 1 p.30, 840-841:

"Relative time refers to normalization of time to the occurrence of the actual time of US arrival".

7. Why in Figure 1B the peak of the red curve is higher than the peak of the black curve, while in Figure 1C both curves have the same height? Wasn't the normalization performed only in the time axis? Confusing...

This is simply due to the fact that in Figure 1C, individual curves were normalized to their respective maximal value in order to assess the scalar property without the confound of peak rate differences. We indeed omitted to stipulate this in the figure legend and have now corrected this mistake p.30, 841-843: **"To assess the scalar property without confound of peak rate differences, both curves were also normalized to their respective maximal values."**

We thank the reviewer for spotting this omission. Doing so, we realized that it would be more accurate to perform the LFP statistics the same way, i.e. assessing the effect of time on data normalized for both x and y axes. This new calculation did not change the outcome of the study. The new values are now provided in Supplementary Table 2.

8. I missed some sort of statistics for proving that the results really follow the scalar property (by the way, the authors may want to define somewhere in the paper what is meant by the scalar property). Computing a single eta squared for the average curves does not seem very convincing to me. Ideally one should test for this scalar property on an individual subject basis... what about performing some sort of correlation or cross-correlation within animals? Or else, perhaps the authors could show the (paired) distribution of the width of the curves as well as normalized peak times? For instance, the authors wrote "As predicted by the scalar property, the width of the curve was reduced proportionally to the time shift, as there was good superposition of both curves", but there is no stats to back this claim. (And related to my first point above, the authors write "proportionally to the time shift", but only a single value of time shift was investigated.... Ideally this proportionality should be assessed using multiple time shift magnitudes...).

We thank the reviewer for these suggestions. Regrettably, we found that performing analyses on individual curves were difficult because of the high variability of the curves of the LFP data. However, to provide some statistical assessment, we performed an additional analysis of the Pearson correlation coefficient (Supplementary Table 1) which shows that, in the low theta range, positive correlations between the 30 s vs. 10 s curves were only significant after normalization of time, with the highest correlation being observed for the coherence. We have now implemented the manuscript with this new analysis.

Furthermore, we have now defined the scalar property in the introduction (p. 3, l.75-76): **"temporal precision proportional to the timed interval"**

We have also rephrased our statements regarding the scalar property as suggested by the reviewer p.5-6, l.130-138:

"There was good superposition of the pre- and post-shift suppression curves when plotted on normalized axes (high η^2 value, an index of superposition, $\eta^2=0.920$, and no before vs. after shift, time X session interaction, $F_{19,95}=1.06$, $p=0.40$), as predicted by the scalar property of interval

timing (Fig.1C). Furthermore, calculation of the Pearson correlation coefficient showed that positive correlations between the 30 s vs. 10 s curves were only significant after normalization of time (Supplementary Table 1). Thus, the shift of peak time was accompanied by a corresponding change in the width of the suppression curves, in agreement with the scalar property."

p.8, l.204-211:

"Comparison of the superposition index (η^2) between averaged curves indicated that the highest superposition was observed for 3-6 Hz coherence (Fig. 4C) as compared to the PSD frequency bands (Fig. 4A-F). Furthermore, calculation of the Pearson correlation coefficient showed that, for the low theta, positive correlations between the 30 s vs. 10 s curves were only significant after normalization of time, with the highest correlation being observed for coherence (Supplementary Table 1). Altogether, these results indicate that BA-DMS interactions are involved in processing the CS-US time interval in aversive Pavlovian paradigms."

We also replaced the word "proportionally" in the following sentence:

p.5, l.129-130:

"Once the behavior had adapted to the new CS-US interval, the width of the suppression curve was also reduced accordingly."

9. The authors could show some examples of non-normalized PSDs. As a reader, I would be interested to know whether the power spectrum exhibits any power peak. And similarly for the coherence spectrum.

We agree with the reviewer and now provide in Supplementary Figure 1 non-normalized theta and gamma PSD for both striatum and amygdala and for coherence spectrum between the two structures when the CS-US interval is set at 30 s. In this figure, we also provide the mean PSDs and coherence across frequencies during pre-stimulus baseline to show stability of the recordings during both the 30 s and 10 s recording conditions.

"Supplementary Figure 1: LFP baseline stability. A-C: Mean DMS and BA PSDs and coherence across frequencies during pre-stimulus baseline showing stability of the recordings for both the 30 s and 10 s conditions. D-F: Examples of non-normalized theta (left) and gamma (right) PSD for both striatum and amygdala, and coherence spectrum between the two structures when the CS-US interval was set at 30 s. BA: Basolateral amygdala; COH: coherence; DMS: dorsomedial striatum; PSD: power spectrum density."

10. I would also be interested to know whether the reported effects are specific for theta and gamma frequency ranges; since the authors show spectra for 0 to ~15 Hz and ~55 to ~95 Hz, one naturally wonders what is happening to the other frequencies.

We understand the reviewer's wonder. However, we specifically chose the theta and gamma bands as we are, for the matter of the current study, interested in plasticity mechanisms typically involving

these frequency ranges. Stimulations in the theta and gamma bands have indeed been reported to trigger synaptic plasticity at cortico-striatal synapses (Charpier and Deniau, 1997; Charpier et al., 1999; Spencer and Murphy, 2000) and this was also confirmed in our hands (Höhn et al., 2011). Given that these frequencies indeed responded to temporal expectancy and are the most relevant to the question addressed in the manuscript, we feel that the extensive and time-consuming process of analyzing all other frequencies is not necessary, and may even divert the focus of the current manuscript.

Charpier, S., Mahon, S. & Deniau, J. M. In vivo induction of striatal long-term potentiation by low-frequency stimulation of the cerebral cortex. Neuroscience 91, 1209–1222 (1999).

Charpier, S. & Deniau, J. M. In vivo activity-dependent plasticity at cortico-striatal connections: evidence for physiological long-term potentiation. Proc. Natl. Acad. Sci. U. S. A. 94, 7036–40 (1997).

Höhn, S. et al. Behavioral and in vivo electrophysiological evidence for presymptomatic alteration of prefrontostriatal processing in the transgenic rat model for huntington disease. J. Neurosci. 31, 8986–97 (2011).

Spencer, J. P. & Murphy, K. P. S. J. Bi-directional changes in synaptic plasticity induced at corticostriatal synapses in vitro. Exp Brain Res 135, 497–503 (2000).

11. *The authors measured arc expression 30, 60, 90 and 150 minutes after the behavioral protocol in "shift" animals. However, the authors only report results for the 90 min group. What about the other time points? In short, the authors should display the same bar graphs as in Figure 5C (which was done for the control animals) for the "shift" animals as well.*

This is indeed a sensible suggestion. We have now incorporated the other time points of Arc immunostaining for the shift group to Figure 5C p.33, l.929-933.

"C: Quantification of the number of striatal Arc-positive cells (mean + SEM) for non-shifted (left panel) or shifted (right panel) animals perfused at different time points after the behavioral session revealed that Arc expression 90min post-training was higher in the DMS than in the DLS for the non-shifted animals."

12. *Related to the point above, the writing gives the impression of a reduced arc expression in "shift" animals, but in reality what is (likely) happening is that arc expression is actually increased, but to a lesser extent than in controls.*

We utterly agree with the interpretation of the reviewer that Arc expression is still upregulated in the DMS of shifted animals, but to a lesser extent than in non-shifted rats. This interpretation was also explicit in our original discussion p.15, l.394-397: **"In accordance with this interpretation, Arc expression was less increased in the DMS (a region specificity previously implicated in several reports^{21,22}) in animals in which reinforcement time was shifted to 10 seconds, suggesting that synaptic strengthening mechanisms were less prominent because of the shift in duration"**

This is actually further supported by the additional Arc immunostaining experiment we provide in Figure 7A-C of the revised manuscript showing that rats that received no CS before perfusion display much lower Arc levels than trained shifted animals.

The revised manuscript has been amended accordingly to make this interpretation unambiguous.

p.9, l.240-243:

"Such a decrease in Arc up-regulation in the DMS suggest that plasticity mechanisms that are continuously taking place due to CS-US presentations are down-regulated as a result of replacement of the 30 s interval by a new duration."

p.15, l.403-406:

"Such view would be in agreement with our observation that the shift to a new CS-US interval is still associated with an up-regulation of Arc, but to a lesser extent than Arc expression triggered by the old duration."

13. In their text, the authors try to relate the reduced arc expression to LTD; is there any evidence for this? A quick search in the literature (and admittedly non-exhaustive) revealed that arc expression may actually be required for LTD (Waung et al, Neuron 2008; reference 36 cited by the authors also show a transient increase)..

The role of Arc in long-term depression (LTD) and long-term potentiation (LTP) is complex as expression of this immediate early gene has been found to be necessary in both forms of synaptic plasticity. Interestingly, whilst LTP would require sustained Arc expression (Bramham et al., 2010; Yilmaz-Rastoder et al., 2011), LTD would be associated with a transient reduction in Arc transcription, followed by an increase (Waung et al., 2008; Yilmaz-Rastoder et al., 2011). Although the precise functions of Arc in synaptic plasticity represent a conundrum still under debate, a consensual view is that different levels of Arc expression would allow for different synaptic changes by sliding the frequency threshold for strengthening or weakening of synaptic efficacy, as depicted in the BCM model of synaptic plasticity (Shepherd and Bear, 2011). Such view would be in agreement with our observation that the shift in CS-US interval is still associated with an upregulation of Arc, but to a lesser extent than Arc expression triggered by the old duration. In this interpretation frame, a lower but still substantial amount of Arc expression would allow for a weakening of synaptic transmission.

We thank the reviewer for these remarks as we have now improved the discussion of the revised manuscript with these considerations p.15, l.397-407:

"Interestingly, whilst LTP would require sustained Arc expression^{41,42}, LTD has been associated with a transient reduction in Arc transcription, followed by an increase^{42,43}. Although the precise functions of Arc in neural plasticity are complex, a consensual view suggests that different levels of Arc expression would allow for different synaptic changes by sliding the frequency threshold for strengthening or weakening of synaptic efficacy, as depicted in the BCM model of synaptic plasticity⁴⁴. Such view would be in agreement with our observation that the shift to a new CS-US interval is still associated with an up-regulation of Arc, but to a lesser extent than Arc expression

triggered by the old duration. In this interpretation frame, a lower but still substantial amount of Arc expression would allow for a weakening of synaptic transmission."

Bramham, C. R. et al. The Arc of synaptic memory. Exp. brain Res. 200, 125–40 (2010).

Shepherd, J. D. & Bear, M. F. New views of Arc, a master regulator of synaptic plasticity. Nat. Neurosci. 14, 279–84 (2011).

Waung, M. W., Pfeiffer, B. E., Nosyreva, E. D., Ronesi, J. A. & Huber, K. M. Rapid translation of Arc/Arg3.1 selectively mediates mGluR-dependent LTD through persistent increases in AMPAR endocytosis rate. Neuron 59, 84–97 (2008).

Yilmaz-Rastoder, E., Miyamae, T., Braun, A. E. & Thiels, E. LTP- and LTD-inducing stimulations cause opposite changes in arc/arg3.1 mRNA level in hippocampal area CA1 in vivo. Hippocampus 21, 1290–301 (2011).

14. The authors state "As down-regulation of Arc expression after the US shift implies a change in synaptic efficacy in the DMS, but not in the DLS, (...) we tested for plasticity at prefronto-DMS synapses induced by the shift of the CS-US interval." Ideally, if the authors want to prove that the change in arc expression is associated to a specific change in synaptic efficacy in DMS but not DLS, why not making the natural control of also recording evoked responses in DLS?

This could have been indeed another possible control. However, the prefronto-DLS connection is anatomically weaker such that in our hands stimulation of the prelimbic cortex does not reliably evoke a stable field potential (EFP) in the DLS (unpublished observation). Inputs to the DLS typically originate from motor areas, but motor cortex-DLS EFPs have not yet been characterized and would potentially be unstable as most likely influenced by movements of the animal. Further, the BA does not project to the DLS. Thus, as the immunohistochemistry results showed no change in DLS related to the temporal shift, we decided not to invest our effort in this technically challenging supplementary control.

15. There is a parenthesis missing in the formula of the suppression index (pg13-14).

We have now corrected this mistake.

16. In the methods, the authors wrote: "Importantly, LFPs were also recorded in alternating sessions in which the lever was removed from the chamber, in order to be able to dissociate electrophysiological activity due to motor control from electrophysiological activity related to timing processes." ... I got confused with the statement of "were also recorded" and caught myself wondering which sessions were actually used in the analysis shown in the paper... was there a difference between recordings in the presence vs absence of the lever?

The LFP recordings shown in the paper were performed only on lever-off sessions in order to avoid electrophysiological activity due to motor control as opposed to activity related to timing processes.

We re-wrote this sentence in order to avoid ambiguity p.19-20, l.518-521:

"Importantly, although animals were always connected to a recording cable, only alternating sessions in which the lever was removed from the chamber were considered for LFP analyses, in order to avoid electrophysiological activity due to motor control as opposed to activity related to timing processes."

We also added this information in the legend of Figure 1 (p. 30 l.834-835): **"Recordings during CS alone trials were analyzed for interleaved sessions without access to lever."**

17. For computing coherence, one needs multiple time windows to average FFT vectors. What was the time window used in this study? I understand that the coherogram was obtained using 5-s sliding windows, but for computing the coherence spectrum of each 5-s window, this should be further divided into smaller windows...

For calculating the PSD and the coherence, we are using an adaptive, weighted multi-taper method. The method involves multiplying orthogonal data windows with the data series (the 5 s interval), computing the FFT and then calculating the weighted average of the direct-spectrum estimates. In other words, the multitaper method used does not sub-divide the time series into length segments. Furthermore, we do not apply zero padding before the FFT.

We have added some clarifications on the multi-taper method description in the Methods and the reference to (Thomson 1982) where this method has been originally proposed. We now specify the Python package we use to compute PSD and coherence using the multitaper method.

p.22, l.593-600:

"This was done by estimating the discrete prolate spheroidal sequences (orthogonal data windows), multiplying each of the tapers with the data series, compute the FFT, and using the adaptive scheme for a better estimation of the discrete-spectrum weighted average⁵⁹. The coherence between LFP signals was computed from the FFT and the weights of the multitaper spectrum estimation⁶⁰. The analysis parameters of the multitaper method were as follows: time-bandwidth product = 3.5, number of used tapers = 7. The PSD and COH calculations are performed using mtspec which is a Python wrapper for the Multitaper Spectrum Estimation Library⁶⁰."

18. Could the authors better explain the permutation test used to check the stats of the cluster analysis? In particular, what random sign flips means?

Random sign flips means that the sign (+1 or -1) is flipped randomly for each data instance; i.e. a random list of +1 and -1 is generated and multiplied to the each data instance. If the mean of the data is different from zero, such a procedure would alter the mean once the random sign flip to each data instance has been applied. Conversely, if the mean is zero, the random sign flip would not alter the mean. So if we observe a mean that is bigger than obtained when permuting, then we are testing non-parametrically if the distribution is zero mean. We have added this explanation to the relevant section in the Methods. p.23, l.618-622:

"Randomized data were generated with random sign flips, i.e., if the data distribution on null hypothesis has zero mean, a random sign flip does not alter the mean. Thus permuting enables to test non-parametrically the null hypothesis. Non-parametric cluster tests were performed using the python implementation of the MNE software package⁶¹."

19. I find a bit odd that the closing sentence of the abstract focuses on the amygdala, while several of the main findings (changes in power, evoked response, Arc+ cells) were obtained for the striatum. Also, there is a punctuation mark (.) missing before this sentence.

We agree that this was true for the previous version of the manuscript. However, the new data we bring in this revised version do strengthen the paper on the control the amygdala exerts on striatal plasticity. We have nevertheless reformulated the closing sentence of the abstract and the title of the manuscript in order to reflect more accurately our findings.

p.1, l.1-2:

"Updating Temporal Expectancy of an Aversive Event Engages Striatal Plasticity under Amygdala Control"

p.2, l.38-40:

"Collectively, this study reveals physiological correlates of plasticity mechanisms of interval timing that take place in the striatum and are regulated by the amygdala."

20. In Figure 4, the abbreviation BEH is not introduced.

The abbreviation BEH has now been introduced in the legend of Figure 4.

21. In Figure 2F bottom, there is a " $p < 0.1$ " which I believe may be a typo.

For all the statistics, the alpha threshold was set at 0.05. However, as the non-parametric cluster analysis was used to narrow the extent of the frequency range considered for PSD and coherence analysis, in order to avoid focusing on excessively narrow bands, the alpha level was set at 0.1 when significant changes were appearing for an alpha of 0.05 in only very small spots, in order to provide us with a band range of tendencies. It is important to differentiate this non-parametric cluster analysis from the ANOVA analyses performed on the selected bands, which tested for interaction between condition (US@30s vs. US@10s) and elapsed time with an alpha value classically set as $p < 0.05$.

Reviewer #4 (Remarks to the Author):

The manuscript by Dallerac et al studies coherence between striatum and amygdala during aversive conditioning, with particular emphasis on the learning that occurs when the intervals between the CS

and US are shifted. The key findings are that changes in the coherence (local field potentials) between amygdala and striatum were correlated with interval changes, and that such changes also correlated with changes in corticostriatal plasticity.

The manuscript is an interesting one, it is novel, it addresses a neglected question, and it reports an enormous amount of time consuming work. I congratulate the authors on their persistence.

Upon my reading the data were largely handled appropriately (but see below re suppression ratio and preCS rates), appropriate credit is assigned to previous work, and the text is clear and well written.

I had the following comments on the experiments and interpretation.

1. The key manipulation here is shifting the delivery of the shock reinforcer, because this reveals the presence of timing at the behavioral level and its neural correlates in amygdala and striatum. I did not understand why the authors only shifted backwards - from 30s CS - US interval to a 10 s interval. The theories they are testing predict that the same changes should occur in animals shifted from 10s to 30s. The point here is that shifting backwards causes a net reduction in overall suppression and presumably fear, and hence it is unclear whether the changes in neural synchronization are due to changes in timing or simply to reductions in fear. If these changes in synchronization were due to learning the new intervals, as the authors and models predict, then they should be observed in animals shifted from 10s - 30s. It may be that the authors did this and I missed it.; if not, it would certainly merit close attention in a discussion.

There was a global reduction of suppression over the entire 60 s CS duration, but equivalent level of maximal suppression for 30 s and 10 s conditions when animals were trained over several weeks after the shift and recorded. However, there was no difference during the shift session, and thus the differential Arc reactivity could not have been due to different global behavioral outputs, as explained in response to reviewer 2 (point#1). Besides, we would like to point out that the potential reduction in fear the reviewer suggests could not explain the LFP temporal pattern we demonstrate and the scalar property. Thus, we have now provided more detailed behavioral data for the Arc labelling experiments, as this is a critical point.

p.9, l.235-240:

"Importantly, as there was no global difference in lever-pressing behavior between these two 90-min subgroups of animals (neither in pre-CS mean lever-press per second 1.29 ± 0.21 vs. 0.95 ± 0.18 , nor during the 60 s CS, mean suppression 0.53 ± 0.02 vs. 0.56 ± 0.03 , maximum suppression 0.87 ± 0.07 vs. 0.88 ± 0.07 ; all $P_s > 0.05$), the difference in Arc labelling could not be related to an unspecific modification in general behavior output, but rather to the detection of new temporal contingencies."

We do agree with the reviewer that the models predict that similar changes should be observed when learning a new duration (i.e. going from 30 s to 10 s, or reverse). In terms of our LFP this would indeed be the case, as the recordings were made after several sessions, i.e. at behavioral stability. However, in our particular paradigm for which the duration of the CS remains the same (i.e. 60 s) and extends beyond the US time of arrival, the learning of the new duration is accompanied with the

extinction of the old duration. Thus, as both processes occur in parallel when the CS-US interval is shifted, the plasticity we observe in the DMS might be the result of one or the other phenomenon, or both. Interestingly though, the fact that inactivating the amygdala blocks such plasticity whilst facilitating extinction (see response to reviewer 1 point #6), raises the possibility that this LTD-like plasticity actually underlies maintenance of the old time interval and competes with a continuous plastic process subserving learning of the new duration. If true, one would predict similar changes regardless of the direction of the shift (long to short vs. short to long) in our paradigm, but not in a situation in which the US is co-terminating with the CS as it renders the two situations asymmetrical (i.e. shifting from 10 s to 30 s engages both learning 30 s and extinguishing 10 s, whereas shifting from 30 s to 10 s engages only the learning of 10 s). This raises interesting possibilities and it would be of interest to see whether the combined changes related to learning a new duration and extinction of the old duration differ depending on whether or not the extinguished duration is embedded within the new expected duration. However, to be meaningful, a complete new dataset including a 10 s to 30 s shift, as well as a comparison with situations in which the CS and US co-terminate, would have to be performed while encompassing all levels of analysis provided in the manuscript, that is, 1/ Behavioral analysis and assessment of the scalar property, 2/ LFP oscillation analyses during behavior, 3/ Arc immunostaining after the shift in an independent experiment, 4/ EFP recording before and after the shift in another independent behavioral experiment, 5/ analysis of the behavioral consequences of inactivating the amygdala and 6/ analysis of the effect of inactivating the amygdala on striatal plasticity. Given the extensive time required to train rats to our timing dedicated auditory aversive conditioning experimental paradigm and the time-consuming analyses involved, we think that doubling the dataset with an opposite shift in CS-US time interval would be unreasonable. Following the reviewer's suggestion, however, we have now mentioned this issue in the discussion p.16, l.416-432:

" An important aspect to consider is the particular double-value the CS acquires in our paradigm, i.e. US predictor before the US time and safety (no-US) value after the US time, which results in the superposition of new learning for the 10 s duration and extinction for the old 30 s duration. Indeed, as both processes occur in parallel, the plasticity we observe at PFC-DMS synapses might be the result of either phenomenon, or both. Interestingly though, the fact that inactivating the amygdala blocks such plasticity whilst facilitating extinction, raises the possibility that this LTD-like plasticity actually underlies maintenance of the old time interval and competes with a continuous plastic process subserving learning the new duration. If true, one would predict similar changes regardless of the direction of the shift (long to short vs. short to long) in our paradigm, but not in a situation in which the US is co-terminating with the CS as it renders the two situations asymmetrical. In any case, the current set of data provides the first compelling evidence, at both the physiological and molecular levels, of the involvement of amygdala-dependent synaptic plasticity mechanisms in the DMS when learning new durations and extinguishing old ones, in agreement with the SBF, the foremost model in interval timing."

2. The authors present 'timing' as fundamental to aversive learning, and of course they are probably correct. But, they used a very large number of training sessions to reveal such evidence for timing. This is not problematic - in fact it is typical of the timing literature. However, it does raise the question of the primacy and functional significance of timing in the aversive system. That is, animals

express evidence for fear/aversive/threat conditioning many trials before they express evidence for timing of the conditioned response. I think the authors could do a better job of addressing this. It does not reduce the significance of what they have reported, but it is theoretically and functionally important to address (how many prey species have this much training with predation? Is this timing a 'secondary' system fine tuning defensive behavior?).

We thank the reviewer for these suggestions. There are indeed several data (including our own) that indicate that timing the CS-US interval is very rapid, and in some cases even after a single trial (Díaz-Mataix et al., 2013), but the full behavioral expression takes a large number of trials, as noted by the reviewer. It raises the question of whether some of the LFP correlates would be present earlier, i.e. at the outset of learning. This is an interesting issue. Some of the mechanisms uncovered in the present manuscript could be at play early in training while others may subtend the flexible and fast adaptation to a sudden change in temporal contingency. The latter could have functional values in terms of flexible behavior and adaptation to changes in the environment. We have now modified the discussion with these added considerations. p.17, l.445-454:

"Nevertheless, it raises the interesting question of whether the exact same neurophysiological/plasticity correlates would be detectable at the outset of learning, as our experiment targets the acquisition of a new duration, and its resulting behavioral adaptation, after overtraining. Noticeably, we previously observed changes in plasticity markers in the basolateral amygdala when shifting the CS-US interval after a single conditioning session¹³, suggesting the involvement of a common network. It however remains possible that only a subset of the correlates we observed here may underlie the flexible and fast adaptation to changes in temporal contingency. Further investigation would therefore be needed to decipher which networks and mechanisms subtend learning of interval, on the one hand, and flexible behavior on the other."

Díaz-Mataix, L., Ruiz Martinez, R. C., Schafe, G. E., LeDoux, J. E. & Doyère, V. Detection of a temporal error triggers reconsolidation of amygdala-dependent memories. Curr. Biol. 23, 467–72 (2013).

3. The ARC data are interesting, but suffer from the absence of a control condition. We have ARC data from shifted and non-shifted rats and the groups differ. This difference is interesting and important, but it is difficult to interpret. Which group is up and which is down?

This is indeed a sensible remark. We now provide in Figure 7C a control condition in which rats are not exposed to the CS before the brain extraction. The data reveal a low basal level of cells expressing Arc, thus indicating that both trained groups in Figure 5 and 7 show upregulation of Arc, but to a lesser extent in the shifted group.

p.34, l.959-960:

"Furthermore, control rats (n=5) not exposed to the CS showed markedly low levels of Arc in both DMS and DLS."

4. *My final substantive comment is that it would be very useful to see some kind of causal manipulation of these circuits on the timing behaviours to determine if they really are critical to the timing. In the absence of such causal data, we are looking at correlations, albeit very interesting ones.*

As explained in response to reviewers 1 & 2, we agree that an interventional experiment would provide more compelling data to support our hypothesis. We have therefore performed additional experimentation whereby we inactivated the amygdala during shift sessions, using the sodium channels blocker lidocaine, and performed an immunostaining for Arc in the DMS and DLS in order to assess striatal plasticity processes. In accordance with our hypothesis that the amygdala facilitates DMS plasticity induced by a change in CS-US interval, the diminution of Arc activation we report in Figure 5D, that we replicate here in Figure 7C, and corroborating the LTD-like plasticity we observed, is blocked by inactivation of the amygdala with lidocaine the day of the shift. These data indicate that blocking the amygdala prevents the change in striatal plasticity processing occurring during learning of a new duration and/or extinguishing the old duration.

To further assess the role of the amygdala on striatal plasticity and duration learning, we have also performed an additional behavioral experiment where activity of the amygdala is inhibited using lidocaine during initial sessions of shift in order to assess whether such inactivation results in a delay in the behavioral adaptation to the new CS-US contingency. Analysis of the evolution of the suppression curve upon the 5 first 2-session blocks shows a significant time X group X blocks interaction when the analysis is restricted to the first 10 s, i.e. the new CS-US interval. The group receiving lidocaine indeed shows a significant interaction time X blocks while the control group does not, indicating a delay in stabilizing the new suppression behavior at CS onset in amygdala inactivated animals. We further analyzed these data by individually fitting suppression curves of each rat for each session block and, using a peakfit analysis, determined the evolution of the behavioral peak time as well as the width of the curve. Surprisingly, such analysis revealed that inactivation of the amygdala with lidocaine does not delay learning of the new 10 s peak time. However, through analyses of the curve's width and start-stop suppression behavior (Tallot et al., 2016), we unraveled differences between the saline vs. lidocaine group, indicating that processes underlying extinction of the 30 s target are compromised by inhibition of the amygdala. Inactivation of the amygdala during the initial error detection phase therefore appears to facilitate extinction of the old 30 s duration. Insofar as the behavioral readout of the rat's temporal expectancy of the new CS-US interval is a function of both learning this new duration and extinguishing the old one, evolution of the suppression peak amplitude would thus, in fact, represents facilitation of the old duration extinction. Taking into account the blockade of striatal plasticity by lidocaine inhibition of the amygdala, a conspicuous interpretation of these data would be that facilitation of PFC-DMS plasticity by the amygdala prevents the extinction of acquired durations and thereby helps to maintain duration memories.

As a whole, these interventional data indicate that the amygdala influences striatal plasticity and thereby tunes duration learning and memory. Thus, the present set of data also provides mechanistic insights on how emotions could influence time estimation.

We have now implemented the manuscript with these new interventional data and their relevance in the result section p.10-13, l. 261-336.

Tallot, L., Capela, D., Brown, B. L. & Doyère, V. Individual trial analysis evidences clock and non-clock based conditioned suppression behaviors in rats. *Behav. Processes* 124, 97–107 (2016).

These substantive points deserve some consideration in the manuscript. These are long and time consuming experiments - the manuscript is interesting and novel as it stands. It may be that addressing most of these issues in the text will be sufficient.

Minor points

1. The authors use 'fear', 'threat', and 'aversive' interchangeably. I would prefer to see one term (ideally aversive or fear) used throughout. It is confusing for readers not expert in the area, or who do not understand the provenance of these terms, to see these different terms.

We agree with the reviewer and now refer to the conditioning paradigm we used as "aversive conditioning".

2. The suppression ratios were calculated in an unusual way. The standard way of calculating a suppression ratio is $CS/(CS - preCS)$. This is the approach that has been used for nearly 50 years. Does the way these ratios are calculated influence the evidence?

The standard way introduced by Annau & Kamin in 1961 is indeed $CS/(CS + preCS)$ (Annau and Kamin, 1961). We just used 1 – this formula to obtain upward curves and used either preCS or averaged ITI for each session (which gives equivalent results), depending on the experimental setup that was used for each experiment. We have replicated the temporal pattern of suppression in our paradigm (Tallot et al., 2016).

Annau, Z. & Kamin, L. J. The conditioned emotional response as a function of intensity of the US. *J. Comp. Physiol. Psychol.* 54, 428–32 (1961).

Tallot, L., Capela, D., Brown, B. L. & Doyère, V. Individual trial analysis evidences clock and non-clock based conditioned suppression behaviors in rats. *Behav. Processes* 124, 97–107 (2016).

3. Finally, it is important to report preCS lever pressing rates as suppression ratios are heavily dependent on these.

PreCS values are important if one want to compare absolute suppression, but not for comparison of its temporal pattern, on which we are focusing on here. Thus, we have now reported the preCS lever pressing values in the text for the two experiments for which it may have impacted the results (i.e. Arc labelling and amygdala inactivation). Importantly, for each specific comparison, there was no difference between groups or conditions, making the suppression ratio a valid measure.

p.9, l.235-238:

"Importantly, as there was no global difference in lever-pressing behavior between these two 90-min subgroups of animals (neither in pre-CS mean lever-press per second 1.29 ± 0.21 vs. 0.95 ± 0.18 , nor during the 60 s CS, mean suppression 0.53 ± 0.02 vs. 0.56 ± 0.03 , maximum suppression 0.87 ± 0.07 vs. 0.88 ± 0.07 ; all $P_s > 0.05$)"

p.12, l.301-303:

"pre-CS lever-pressing level remained stable (from 1.60 ± 0.16 to 1.38 ± 0.13 lever-press per second for lidocaine, and 1.39 ± 0.11 to 1.23 ± 0.12 for saline)".

REVIEWERS' COMMENTS:

Reviewer #1 (Remarks to the Author):

The authors have been highly responsive to the reviewers comments. Not only have they collected new data to extend and support their findings, but they have also incorporated additional analyses and alternative explanations into their Methods and Discussion sections. Overall, this is a much improved manuscript address a cutting-edge topic in the fields of learning, memory, and interval timing.

Reviewer #2 (Remarks to the Author):

This paper improved by consideration of potential differences in behavior among the groups tested, and by the addition of the lidocaine inactivation experiments. In fact, the lidocaine behavioral results are surprising: learning the new timing interval was actually faster after amygdala inactivation. The authors come up with a somewhat complex interpretation in which amygdala projections to the striatum facilitate LTP-like plasticity that maintains 30 sec timing, so that when this is blocked, the 30 sec timing behavior extinguishes faster in favor of 10 sec timing behavior. Although their data is consistent with this interpretation, the study does not conclusively demonstrate that this mechanism is in effect. The authors should make this more clear, and discuss alternative interpretations more fully. In addition, because their developing model is complex, I think it would be helpful to include a diagram describing their interpretation and perhaps one or two competing alternatives.

Reviewer #3 (Remarks to the Author):

The authors have satisfactorily addressed my concerns; I have no further objection against the publication of this manuscript.

Reviewer #4 (Remarks to the Author):

The authors have extensively revised their manuscript and addressed all of my comments, including the addition of a new data reporting the effects of amygdala inactivation. I would have preferred to see a more precise manipulation using either a chemogenetic or optogenetic approach, but the new data do strengthen the manuscript and strengthen the authors conclusions.

Responses to referees

Reviewer #1 (Remarks to the Author):

The authors have been highly responsive to the reviewers comments. Not only have they collected new data to extend and support their findings, but they have also incorporated additional analyses and alternative explanations into their Methods and Discussion sections. Overall, this is a much improved manuscript address a cutting-edge topic in the fields of learning, memory, and interval timing.

We thank the reviewer for acknowledging we took her/his advices into account.

Reviewer #2 (Remarks to the Author):

This paper improved by consideration of potential differences in behavior among the groups tested, and by the addition of the lidocaine inactivation experiments. In fact, the lidocaine behavioral results are surprising: learning the new timing interval was actually faster after amygdala inactivation. The authors come up with a somewhat complex interpretation in which amygdala projections to the striatum facilitate LTP-like plasticity that maintains 30 sec timing, so that when this is blocked, the 30 sec timing behavior extinguishes faster in favor of 10 sec timing behavior. Although their data is consistent with this interpretation, the study does not conclusively demonstrate that this mechanism is in effect. The authors should make this more clear, and discuss alternative interpretations more fully. In addition, because their developing model is complex, I think it would be helpful to include a diagram describing their interpretation and perhaps one or two competing alternatives.

We agree with the reviewer that our results bring up several interpretations which are somewhat complex, as often is the case in integrative neurosciences and the biology of interval timing. We feel we have extensively discussed possible plasticity mechanisms associated with our observations that the amygdala influences cortico-striatal plasticity and the learning/extinction of the CS-US interval p14-16:

“With regard to the SBF theory for interval timing, strengthening of specific patterns of cortico-striatal activation are thought to enable striatal memory storage of manifold intervals through reinforcement of cell/synaptic assemblies to be compared with ongoing patterns of cortico-striatal activations⁶. One interpretation of such theory is that multitudinous sets of cortico-striatal synapses are continuously being weighted across time intervals. In our configuration this would imply that the pattern of cortico-striatal synapses corresponding to the 30 seconds time interval was subjected to long-term changes in synaptic strength upon each presentation of the CS-US pairing. Such an interpretation is supported by our immunostaining of the immediate early gene and marker of synaptic plasticity Arc in the DMS and DLS, which showed a typical increase in protein expression 90 minutes after training^{39,40} in animals that remained subjected to the same

CS-US time interval, thus revealing a network undergoing recent synaptic changes. Strikingly, the LTD-like plasticity we observed as a result of a change in timing of the US arrival also supports this view. Indeed, if the 30 seconds interval is constantly being decoded and stored through synaptic reinforcement, a shift to the 10 seconds interval would cause the involved synapses to undergo depotentiation and settle at a different (reduced) synaptic strength. In accordance with this interpretation, Arc expression was less increased in the DMS (a region specificity previously implicated in several reports^{21,22}) in animals in which reinforcement time was shifted to 10 seconds, suggesting that synaptic strengthening mechanisms were less prominent because of the shift in duration. Interestingly, whilst LTP would require sustained Arc expression^{41,42}, LTD has been associated with a transient reduction in Arc transcription, followed by an increase^{42,43}. Although the precise functions of Arc in neural plasticity are complex, it is thought that different levels of Arc expression would allow for different synaptic changes by sliding the frequency threshold for strengthening or weakening of synaptic efficacy, as depicted in the BCM model of synaptic plasticity⁴⁴. Such a view would be in agreement with our observation that the shift to a new CS-US interval is also associated with an up-regulation of Arc, but to a lesser extent than Arc expression triggered by the old duration. By this interpretation, a lower but still substantial amount of Arc expression would allow for a weakening of synaptic transmission. Thus, the LTD-like plasticity we observed might also reflect LTD mechanisms, rather than depotentiation, enabling behavioral adaptation to the new CS-US interval. It is worth mentioning that a recent study showed that infusion of anisomycin, a protein synthesis inhibitor, into the dorsal striatum of rats did not prevent the rapid learning of a new time of reinforcement arrival in an appetitive instrumental peak interval paradigm⁴⁵. Since both pre- and postsynaptic LTD co-exist at cortico-striatal synapses⁴⁶⁻⁵⁰, one possibility is that learning of a new duration also involves presynaptic LTD mechanisms, which may be independent of protein synthesis⁴⁷. Alternatively, such result could also support the view that depotentiation rather than bona fide LTD mechanisms are at play and do not involve gene expression. An important aspect to consider is the particular double-value the CS acquires in our paradigm, i.e. US predictor before the US time and safety (no-US) value after the US time, which results in the superposition of new learning for the 10 s duration and extinction for the old 30 s duration. Indeed, as both processes occur in parallel, the plasticity we observe at PFC-DMS synapses might be the result of either phenomenon, or both. Interestingly though, the fact that inactivating the amygdala blocks such plasticity whilst facilitating extinction, raises the possibility that this LTD-like plasticity actually underlies maintenance of the old time interval and competes with a continuous plastic process subserving learning the new duration.”.

However, we do acknowledge that the possibility of an intermediate structure connecting both the dorso-medial striatum and the amygdala remains and should be mentioned in the discussion. We have therefore amended the manuscript accordingly p.16:

“It also remains possible that the interplay between blockade of plasticity and the facilitation of time interval updating involves another brain structure sharing connections with both the striatum and amygdala”.

Finally, we fully agree with the reviewer that providing a diagram explaining the main interpretation of our results would be welcome. We thus now illustrate our conceptual framework in the diagram provided as Figure 10.

Reviewer #3 (Remarks to the Author):

The authors have satisfactorily addressed my concerns; I have no further objection against the publication of this manuscript.

We thank the reviewer for acknowledging we took her/his advices into account.

Reviewer #4 (Remarks to the Author):

The authors have extensively revised their manuscript and addressed all of my comments, including the addition of a new data reporting the effects of amygdala inactivation. I would have preferred to see a more precise manipulation using either a chemogenetic or optogenetic approach, but the new data do strengthen the manuscript and strengthen the authors conclusions.

We thank the reviewer for acknowledging we took her/his advices into account.